

# New isoprenoid GDGT index as a water mass and temperature proxy in the Southern Ocean

Hana Ishii[1,2,3], Osamu Seki[1,2], Masanobu Yamamoto[2], Bella Duncan[3,4]

[1] Institute of Low Temperature Science, Hokkaido University, Sapporo, 060-0819, Japan
[2] Faculty of Environmental Earth Science, Hokkaido University, Sapporo, 060-0810, Japan
[3] now at Antarctic Research Centre, Victoria University of Wellington, Wellington, 6140, New Zealand
[4] Earth Science New Zealand, Lower Hutt, 5011, New Zealand

*Correspondence to*: Hana Ishii (hana.ishii@vuw.ac.nz), Osamu Seki (seki@lowtem.hokudai.ac.jp)

**Abstract.** The Southern Ocean plays a crucial role in the global carbon cycle, ocean heat transport, and Antarctic ice dynamics. Investigating past variability in the Southern Ocean, including temperature and water masses distribution, can improve understanding of how this system may respond to current climate change. Isoprenoid glycerol dialkyl glycerol tetraethers (isoGDGT) can be used as an ocean temperature proxy and have been applied to sediments in the Southern Ocean to reconstruct past temperature variability. However, applications of current isoGDGT-based temperature indices are subject to substantial uncertainty in the Antarctic Zone. In this study, we propose a new isoGDGT-based index, the so-called Antarctic IsoGDGT Zonal (AIZ) index, as a zonal water mass tracer of the Antarctic Circumpolar Current (ACC) based on reanalysed Southern Ocean core-top data. We also found that the AIZ index exhibits a significant correlation with subsurface temperature (subST) south of the Polar Front, suggesting that it can be used as a temperature proxy in the Antarctic Zone (subST = 24.17 ×AIZ–1.45 ($R^2$ = 0.81, $n$ = 134, $p$ < 0.0001)). Applying the AIZ index to late Pleistocene sediment cores collected around the ACC zone confirms its reliability as a water mass tracer and temperature proxy in the Antarctic Zone. Our study highlights the high potential of isoGDGT for reconstructing palaeoceanographic conditions in the Southern Ocean.

## 1 Introduction

The Southern Ocean contains the world's largest ocean current, the Antarctic Circumpolar Current (ACC), which plays a critical role in regulating global ocean circulation, the carbon cycle, and the stability of the Antarctic ice sheet, and thereby significantly influencing the global climate (Carter et al., 2008; Chapman et al., 2020). A better understanding of the substantial role of the Southern Ocean in the Earth's climate system is therefore considered an important research subject in climate science. Because response times of the oceans and ice sheets to climate change are found to vary over a range of timescales from short-term (sub-daily to decadal) to long-term (multi-millennial) (Hanna et al., 2024; Yang & Zhu, 2011), understanding their role requires the study of long-term climate variability in the geological past. This, in turn, necessitates reliable proxy-





based palaeoclimate records from the Southern Ocean, particularly from high latitude regions such as the Antarctic Zone.
Therefore, it is important to develop and refine palaeoceanographic proxies applicable to this region.

Isoprenoid glycerol dialkyl glycerol tetraethers (isoGDGT) are membrane lipids produced primarily by ammonia-oxidizing
marine Thaumarchaeota as well as the methanogenic Euryarchaeota in the marine environment (Schouten et al., 2013). These
compounds are ubiquitous across the global ocean, including polar regions (Ho et al., 2014; Schouten et al., 2002, 2013) and
are well-preserved in marine sediments (de Bar et al., 2019). IsoGDGT have six different structures with different numbers of
cyclopentane rings (GDGT-0, GDGT-1, GDGT-2, and GDGT-3) and cyclohexane moieties (Crenarchaeol (Cren) and its
regioisomer (Cren')) (Fig. S1). The distribution of isoGDGT is strongly correlated with ocean temperature in global core-top
datasets (Schouten et al., 2002). Based on this relationship, Schouten et al. (2002) proposed the first isoGDGT index ($TEX_{86}$).
Since then, a number of studies have been conducted to assess the proxy, propose new indices and develop new approaches
regarding isoGDGT-based palaeothermometry (e.g., Dunkley Jones et al., 2020; Kim et al., 2010; Tierney and Tingley, 2014).

IsoGDGT have been used to estimate sea surface temperatures (SST) in the Southern Ocean (Fietz et al., 2020; Kim et al.,
2010, 2012; Shevenell et al., 2011), but isoGDGT distributions at high latitudes show considerable variability and a different
relationship to temperature than core-top samples from mid and low latitudes (Fig. S2). The weak correlation between currently
used isoGDGT indices, such as $TEX_{86}$ and $TEX_{86}^{L}$ (Kim et al. 2010), and SST in the Antarctic Zone is likely attributed to non-
thermal factors. These include regional variability in archaeal community composition (Pearson & Ingalls, 2013; Spencer-
Jones et al., 2021), habitat depth of isoGDGT producers (Ho & Laepple, 2016; Jaeschke et al., 2017; Kim et al., 2012; Seki et
al., 2014; Spencer-Jones et al., 2021) and seasonality of isoGDGT production (Chandler & Langebroek, 2021; Church et al.,
2003; Park et al., 2019). These studies highlight that polar-specific factors in isoGDGT production should be taken into account
when investigating the application of isoGDGT as an environmental proxy in these regions.

On the other hand, recent studies have suggested that hydroxylated-GDGT also produced by marine archaea have high potential
as a palaeotemperature proxy applicable in high latitudes (Fietz et al., 2020; Liu et al., 2020; Park et al., 2019; Varma et al.,
2023). However, the utility of isoGDGT proxies in high latitudes has not yet been fully explored, leaving considerable room
for further investigation.

In this study, we reanalyzed isoGDGT in core-top datasets from the south of 35° S to develop a new isoGDGT-based proxy to
further explore the potential use of isoGDGT for reconstructing palaeoceanographic changes in the Southern Ocean. Based on
the statistical reanalysis, we propose the new isoGDGT-based index, so-called the Antarctic IsoGDGT Zonal (AIZ) index,
designed to estimate zonal water mass change in the Southern Ocean and reconstruct temperature south of the Polar Front. The
robustness of this index as a water mass tracer and temperature proxy is evaluated by comparison to previously published and
newly generated sedimentary isoGDGT records covering the past 160 kyr in the Southern Ocean.




## 2. Oceanographic settings

The Southern Ocean encompasses the broad oceanic regions surrounding Antarctica, generally south of approximately 35°S (Chapman et al., 2020). At large scales, the Southern Ocean circulation is dominated by the strong eastward-flowing ACC which connects all three major ocean basins. The ACC plays a critical role in the global distribution of heat, salt, carbon, nutrients and dissolved gases (Carter et al., 2008). It comprises several oceanic fronts characterized by dynamic and complex features influenced by multiple factors such as atmospheric processes, changes in wind strength and belts, and bathymetry
(Chapman et al., 2020). In the upper water column, water properties such as temperature, salinity, oxygen, and nutrients show distinct transitions across these fronts (Chapman et al., 2020). These fronts include the Subtropical Front (STF), Subantarctic Front (SAF), Polar Front (PF), Southern ACC Front (SACCF) and Southern Boundary Front (SBF) from north to south (Fig. 1; Carter et al., 2008; Orsi et al., 1995). The Southern Ocean is meridionally divided into four major zones delimited by these oceanic fronts: the Subantarctic Zone, the Polar Frontal Zone, the Antarctic Zone, and the Zone south of the ACC (Pollard et
al., 2002).

The Subantarctic Zone, north of the SAF, features relatively warm (7–4 °C) and saline (~34.0) surface waters enriched in nutrients, which promote high primary productivity (Carter et al., 2022). The Polar Frontal Zone, located between the SAF and PF serves as a transition area where Subantarctic Surface Water cools and freshens as it moves southward (~4 °C and
salinity 34.0–33.8) (Carter et al., 2022). The Antarctic Zone, extending from the PF to the SACCF, is characterized by a thin (~100 m thick) Antarctic Surface Water layer. In this zone, SST drops below 4 °C with salinity ~34.0 (Carter et al., 2022). The zone south of the ACC, near the Antarctic continental margin, is characterized by continued cooling of Antarctic Surface Water, reaching near-freezing temperatures with a salinity of ~34.0 (Carter et al., 2022). These water mass properties, however, exhibit considerable regional variability across the Southern Ocean (Carter et al., 2022).


The Southern Ocean's unique oceanographic character is further defined by the dynamics of Circumpolar Deep Water (CDW), which forms the most voluminous water mass in the region (Carter et al., 2009). It circulates eastward along the ACC and extends from ~1400 m to >3500 m depth, but south of the PF, CDW upwells to a depth centred at ~500 m and reaches the ocean surface near the continental shelf (Carter et al., 2009; Holland et al., 2020). This is driven by wind-induced upwelling
where westerly winds transition to polar easterly winds in the Antarctic Zone (Carter et al., 2022). These upwelling events bring deep, nutrient-rich waters to the surface, creating strong connectivity between surface and subsurface water masses (Carter et al., 2022).



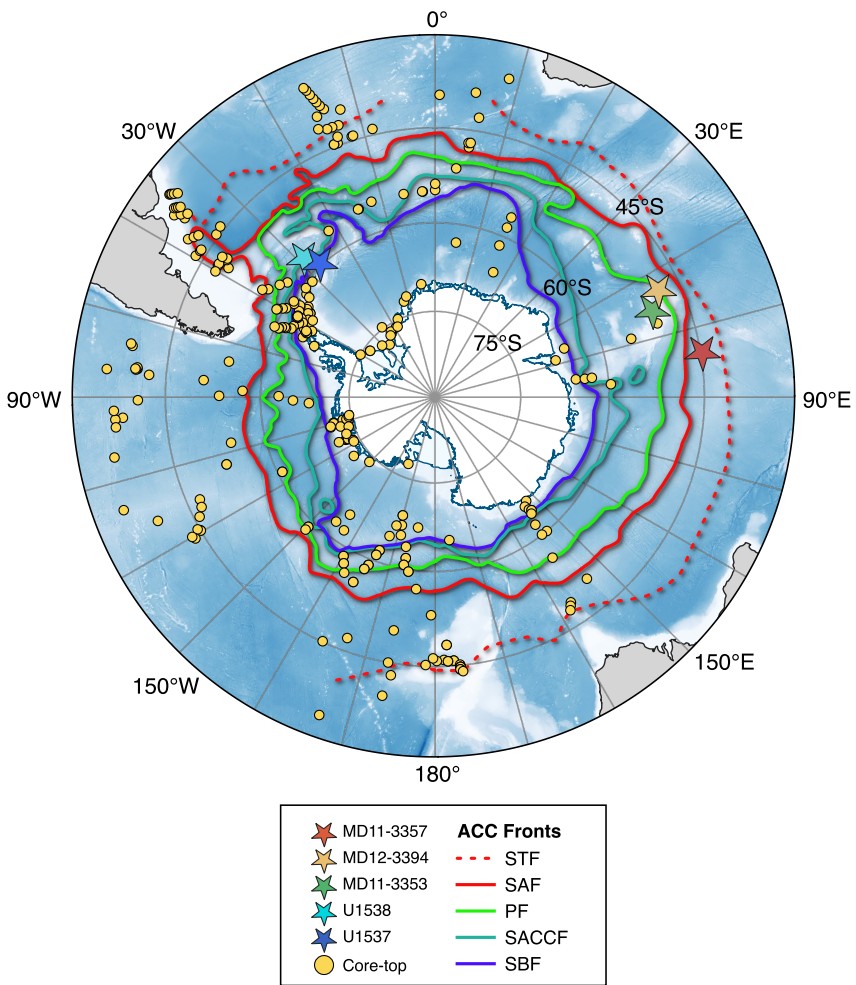

**Figure. 1: Location of the core-top sediments and sediment cores analysed in this study. Core-top samples (yellow circles, *n* = 289) are from Tierney and Tingley (2014), Jaeschke et al. (2017) and Lamping et al. (2021). Sediment cores are shown as stars: dark red (MD11-3357), dark yellow (MD12-3394), green (MD11-3353), light blue (U1538), blue (U1537). Subtropical Front (STF), Subantarctic Front (SAF), Polar Front (PF), Southern ACC Front (SACCF) and Southern Boundary Front (SBF) are also indicated. The oceanographic fronts and bathymetric model are adapted from Orsi et al. (1995) and the GEBCO Bathymetric Compilation Group (2023), respectively. The map was created using Quantarctica (Matsuoka et al., 2018).**

## 3 Materials and Methods

### 3.1 Core-top samples and environmental parameters used in this study

In this study, we reanalysed a global (*n* = 916) and south of 35°S (*n* = 289) core-top isoGDGT dataset, by adding data from Jaeschke et al. (2017) and Lamping et al. (2021) to the calibration dataset presented in Tierney & Tingley (2014) (Fig. 1). SST data for each core-top site were derived from the World Ocean Atlas 2009 (WOA09) 1° × 1° gridded product (Locarnini et al.,



2010). Additional environmental parameters that potentially influence isoGDGT composition—including annual mean and seasonal seawater temperatures (0–1400 m depths), salinity (0–1400 m depth), and oxygen saturation (0–1400 m depth) — were derived from the World Ocean Atlas 2018 (WOA18) 0.25° × 0.25°gridded product (Boyer et al., 2018).

## 3.2 Sediment cores used in this study

Previously reported and newly generated sedimentary isoGDGT records from multiple sites around the ACC were used to
evaluate the applicability of isoGDGT as a palaeoceanographic proxy in the Southern Ocean (Fig. 1). New isoGDGT records were obtained from sediment cores U1538 (57.43° S, 43.35° W, 3131 m water depth) and U1537 (59.11° S, 40.91° W, 3713 m water depth) (Fig. 1) collected in the northern and southern Scotia Sea near the SBF during the International Ocean Drilling Program Expedition 382 (Weber et al., 2019). The sites are located in the Dove Basin in the southern Scotia Sea, where sedimentation is influenced by the transport of eastward-flowing ACC and northward-flowing Weddell Sea Deep Water
(Weber et al., 2019). Additionally, we analysed previously reported isoGDGT records in sediment cores MD11-3357 (44.68° S, 80.43° E, 3349 m water depth), MD12-3394 (48.38° S, 64.58° E, 2320 m water depth) and MD11-3353 (50.57° S, 68.39° E, 1568 m water depth) collected from the southeastern Indian Ocean (Ai et al., 2020; 2024).

## 3.3 GDGT analysis

Organic compounds were extracted from approximately 2 g of freeze-dried sediment using a DIONEX Accelerated Solvent Extractor 200 with dichloromethane/methanol (9:1) at a temperature of 100 °C and a pressure of 1000 psi. The extracts were initially separated into neutral and acidic fractions by aminopropyl silica gel column chromatography with dichloromethane/2-propanol mixture (2:1). The neutral fraction was further separated into two fractions (N1-3 and N4) by silica gel column chromatography. The N4 fraction, which contains GDGT, was dissolved in hexane/2-propanol (99:1) and filtered through 0.45
μm filters for GDGT measurement.

GDGT in U1537 and U1538 samples were identified and quantified using high-performance liquid chromatography-mass spectrometry (HPLC-MS) with an Agilent 1260 HPLC system coupled to 6130 quadrupole mass spectrometers. Separation was achieved with a Prevail Cyano column (2.1 × 150 mm, 3 μm; Grace Discovery Science, USA) maintained at 30 °C
following the method of Hopmans et al. (2000) and Schouten et al. (2007). The analytical conditions were as follows: flow rate 0.2 mL/min, isocratic with 99% hexane and 1% 2-propanol for the first 5 min followed by a linear gradient to 1.8% 2-propanol over 45 min. Detection was achieved with an atmospheric pressure chemical ionization-MS (APCI-MS). The spectrometer was run in two different selected ion monitoring modes (m/z 743.8, 1018, 1020, 1022, 1032, 1034, 1036, 1046, 1048, 1050, 1292.3, 1296.3, 1298.3, 1300.3, and 1302.3). Following the literature of Hopmans et al. (2004), compounds were
identified by comparison of mass spectra and retention times. GDGT were detected and quantified by integrating the peak area





in the $(M+H)^+$ chromatogram with a comparison to the peak area of an internal standard ($C_{46}$ GTGT) in the $(M+H)^+$ chromatogram, according to the method of Huguet et al. (2006). By comparing the peak areas of isolated Cren, GDGT-0 and $C_{46}$ GTGT in known amounts, the correction value of ionization efficiency between GDGT and the internal standard was determined (Schouten et al., 2007). To monitor the changes in the ionization efficiency, a mixture of $C_{46}$ GTGT as the working

standard and the GDGT extracted and purified from an East China Sea sediment was inserted in the routine analysis every 20 samples. The standard deviations of Cren, GDGT-1, GDGT-2, and GDGT-3 in replicate LC/MS analysis were 1, 1, 3, and 2% in sediment samples, respectively.

**3.4 GDGT-based indices used in this study**

A number of indices and calibrations have been proposed to reconstruct ocean temperature and assess non-thermal effects
based on isoGDGT composition. The isoGDGT-based indices and methods used in this study are described below. The $TEX_{86}$ index was first proposed by Schouten et al. (2002) and is defined as the following Eq. (1), where the bracketed GDGT represent the relative abundance of each compound.

$$TEX_{86} = \frac{[GDGT-2]+[GDGT-3]+[Cren']}{[GDGT-1]+[GDGT-2]+[GDGT-3]+[Cren']} \tag{1}$$

The conversion of the $TEX_{86}$ index to temperature was initially achieved using a linear relationship (Schouten et al., 2002). A
range of calibrations have subsequently been developed (Kim et al., 2010; Liu et al., 2009; Schouten et al., 2002). This includes a spatially varying calibration model for $TEX_{86}$ based on a Bayesian approach, BAYSPAR (Tierney & Tingley, 2014). The $TEX_{86}^L$ index, was also proposed as more suitable for reconstructing temperatures below 15 °C and is defined as the following Eq. (2) (Kim et al., 2010; 2012).

$$TEX_{86}^L = \log_{10}\left(\frac{[GDGT-2]}{[GDGT-1]+[GDGT-2]+[GDGT-3]}\right) \tag{2}$$

The OPTiMAL calibration, a machine learning approach, uses all six isoGDGT in global core-top data as training data to estimate SST (Dunkley Jones et al., 2020). The training dataset ($n = 914$) used in this study consists of the global "Op1" dataset (Dunkley Jones et al., 2020) and additional core-top data from the Southern Ocean (Jaeschke et al., 2017; Lamping et al., 2021).

IsoGDGT are biosynthesized not only by Thaumarchaeota but can also be derived from other sources including methanogenic, methanotrophic and terrigenous archaea. A number of isoGDGT-based indices have been proposed to assess the non-thermal effects caused by the exogenous isoGDGT inputs (Blaga et al., 2009; Sinninghe Damsté et al., 2012; Taylor et al., 2013; Pearson & Ingalls, 2013). The methane index (MI) has been proposed as an indicator of post-depositional methanotrophic archaeal GDGT input if the value exceeds 0.3 and is defined as follows (Zhang et al., 2011),

$$MI = \frac{[GDGT-1]+[GDGT-2]+[GDGT-3]}{[GDGT-1]+[GDGT-2]+[GDGT-3]+[Cren]+[Cren']} \tag{3}$$



The relative abundance of Cren' to Cren, expressed as fcren (Eq. (4)) is interpreted as reflecting non-temperature-related influences, particularly community changes in Thaumarchaeota that produce differing amounts of Cren' (O'Brien et al., 2017; Pitcher et al., 2010). This is relevant when fcren exceeds 0.25 (O'Brien et al., 2017):

$$\text{fcren} = \frac{[\text{Cren}']}{[\text{Cren}']+[\text{Cren}]} \tag{4}$$

Additionally, the ratio of GDGT-2 and GDGT-3 ([2]/[3]) has been suggested to reflect a contribution from archaea living deeper in the water column, especially when the values exceed 5 (Taylor et al., 2013).

$$[2]/[3] = \frac{[\text{GDGT}-2]}{[\text{GDGT}-3]} \tag{5}$$

%GDGT-0 has been proposed as an indicator of the contribution of methanogenic archaea when the value exceeds 67 %, defined as follows (Sinninghe Damsté et al., 2012):

$$\%[\text{GDGT}-0] = \frac{[\text{GDGT}-0]}{[\text{GDGT}-0]+[\text{Cren}]} \times 100 \tag{6}$$

## 4 Results and Discussion

### 4.1 Regional features of isoGDGT distributions in the Southern Ocean

As shown in Fig. S2, correlations between the conventional isoGDGT indices ($\text{TEX}_{86}$ and $\text{TEX}_{86}^{L}$) and satellite-derived in situ SST are weak in the low temperature range (< 5 °C), calling into question the applicability of the conventional indices as SST proxies to the polar region.

Therefore, we conducted principal component analysis (PCA) on the fractional abundances of six isoGDGT from sites located south of 35° S ($n$ = 289; Fig. 2a) to evaluate the factors specific to the production of isoGDGT in the Southern Ocean. For comparison, PCA was also performed on a global core-top dataset ($n$ = 916; Fig. 2b). In both the Southern Ocean and global datasets, the first two principal components explain over 90% of the total variance, with PC1 accounting for approximately 70% (Table 1).

In the Southern Ocean dataset, PC1 exhibits strong positive loadings for GDGT-1 (0.69) and GDGT-2 (0.56), suggesting that these two compounds are the primary contributors to this component (Table 1, Fig. 2a). In contrast, GDGT-0 displays a negative loading (-0.23) on PC1, which is also observed in the global PCA. However, the primary contributors to PC1 in the global dataset differ, with GDGT-2 and Cren' showing the highest loadings in contrast to the Southern Ocean dataset (Table 1, Fig. 2b).

For PC2, the Southern Ocean dataset is primarily influenced by GDGT-0 and Cren, which exhibit opposing loadings: GDGT-0 loads negatively, while Cren loads positively. In contrast, global PC2 is mainly driven by GDGT-1 and Cren, which also



load in opposite directions. These differences highlight that the distribution of isoGDGT in the Southern Ocean is not entirely consistent with that observed in the global dataset.

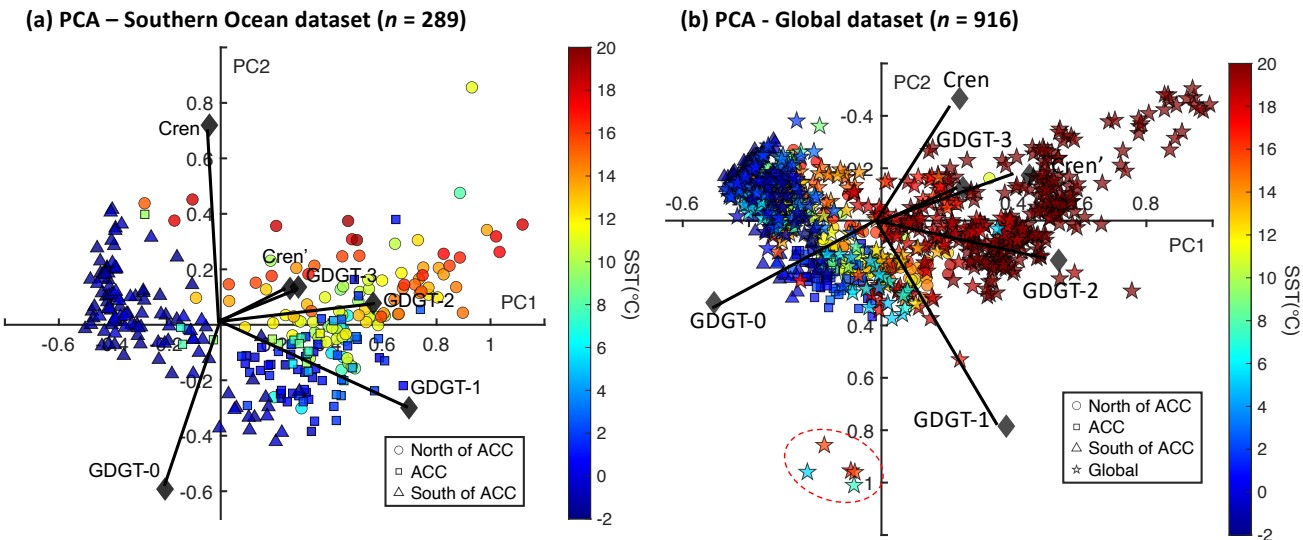

**Figure. 2: PCA biplots for core-top isoGDGT data from (a) the Southern Ocean (*n* = 289) and (b) the global dataset (*n* = 916). The colour bar shows the Sea Surface Temperature (SST) range. The Southern Ocean samples are classified into three groups: north of the ACC (circles), centre of the ACC (squares) and south of the ACC (triangles), separated by the Subantarctic Front and the Southern Boundary of ACC. The global samples are shown as stars. Five sites forming separate clusters (red dotted circles) represent samples that failed quality screening.**

**Table 1: Proportion of variance and loading values for each isoGDGT component from PCA using the Southern Ocean (SO) (*n* = 289) and Global (*n* = 916) datasets.**

|  | Proportion of variance | Loading numbers | | | | | |
|---|---|---|---|---|---|---|---|
|  |  | GDGT-0 | GDGT-1 | GDGT-2 | GDGT-3 | Cren | Cren' |
| PC1 (SO) | 70.3 % | **-0.23** | **0.69** | **0.56** | 0.25 | -0.08 | 0.29 |
| PC2 (SO) | 22.1 % | **-0.53** | -0.32 | 0.13 | 0.17 | **0.74** | 0.15 |
| PC1 (Global) | 69.3 % | **-0.51** | 0.38 | **0.54** | 0.25 | 0.24 | **0.45** |
| PC2 (Global) | 20.9 % | 0.31 | **0.78** | 0.15 | -0.13 | **-0.47** | -0.17 |

The loading numbers with the higher values are shown in bold.

## 4.2 Environment controls on isoGDGT variability: Insights from PCA of core-top isoGDGT

To determine what PC1 represents in both global and Southern Ocean datasets, we compared it with the environmental parameters that potentially influence isoGDGT composition, including SST, salinity, and oxygen saturation at surface (0 m) depth (Qin et al., 2015; Schouten et al., 2013). Five sites in the North Pacific were excluded from the global core-top dataset, as they formed distinct clusters in the PCA and exceeded the thresholds of the screening methods (MI and %GDGT-0) (Fig.





2b). In the global dataset, PC1 exhibits the strongest correlation with SST (Table 2: $R = 0.89$), supporting the use of GDGT

distribution as a palaeotemperature proxy. In the Southern Ocean dataset, PC1 also shows the highest correlation with SST

among the environmental variables; however, the correlation is considerably weaker than in the global dataset (Table 2; $R =$

0.69). This suggests that, in contrast to the global dataset, the isoGDGT distribution in the Southern Ocean may be substantially

influenced not only by temperature but also by other environmental factors.

We further examined relationship of PC1 with non-thermal GDGT indices such as MI (Zhang et al., 2011), fcren (O'Brien et

al., 2017), [2]/[3] (Taylor et al., 2013), and %GDGT-0 (Sinninghe Damsté et al., 2012). Both the Southern Ocean and global

PC1 show strong positive correlations with MI and fcren (Table 2), with values remaining below the critical thresholds of 0.3

and 0.25, respectively. This indicates that anomalous GDGT distributions are not apparent, suggesting the contribution of post-

depositional methanotrophic archaea is negligible.


One of the notable distinctions between the global and Southern Ocean dataset lies in the relation between PC1 and [2]/[3]. As

mentioned in the Methods (Section 3.4), [2]/[3] values above 5 indicate significant contributions from deeper-dwelling archaea.

While global PC1 shows a weak correlation with [2]/[3] (Table 2; $R = 0.38$), Southern Ocean PC1 exhibits a much stronger

correlation (Table2; $R = 0.74$), with several values exceeding this threshold (Fig. S5a). This association implies that isoGDGT

in sediment samples in the Southern Ocean are more strongly influenced by archaea living below the surface water. This

subsurface influence is further confirmed by the positive correlation between [2]/[3] value and water depth at each site ($R =$

0.74; Fig. S5b), indicating that export depth, alongside temperature, plays a key role in shaping the distribution of isoGDGT

in the Southern Ocean. The stronger influence of export depth in the Southern Ocean, compared to the global dataset, could

be explained by the vertical mixing or upwelling of CDW (Church et al., 2003; Kim et al., 2012; Liu et al., 2020; Spencer-

Jones et al., 2021), a key oceanographic feature of the region.

The relationship between PC1 and %GDGT-0 is also markedly different between the global and Southern Ocean datasets. In

the global dataset, %GDGT-0 exhibits a strong negative correlation with PC1 (Table 2; $R = -0.87$), whereas in the Southern

Ocean, this correlation is much weaker (Table 2; $R = -0.25$). Interestingly, the slope of the relationship changes across the PF:

sites south of the PF exhibit a positive slope, while those to the north follow the negative trend observed globally (Fig. 3a; see

also Fig. S4a). A similar divide is evident in the relationship between PC1 and SST in the Southern Ocean (Fig. 3b), where the

data form two distinct clusters corresponding to locations north and south of the PF. In Cluster 1, which primarily represents

sites in the Antarctic Zone (south of the PF), the correlation between PC1 and SST is stronger ($R = 0.77$) than that observed

for the full Southern Ocean dataset ($R = 0.69$). These findings indicate that the distribution of isoGDGT in the Southern Ocean

differs between the regions north and south of the PF, with SST exerting a stronger influence on isoGDGT distribution in the

Antarctic Zone.





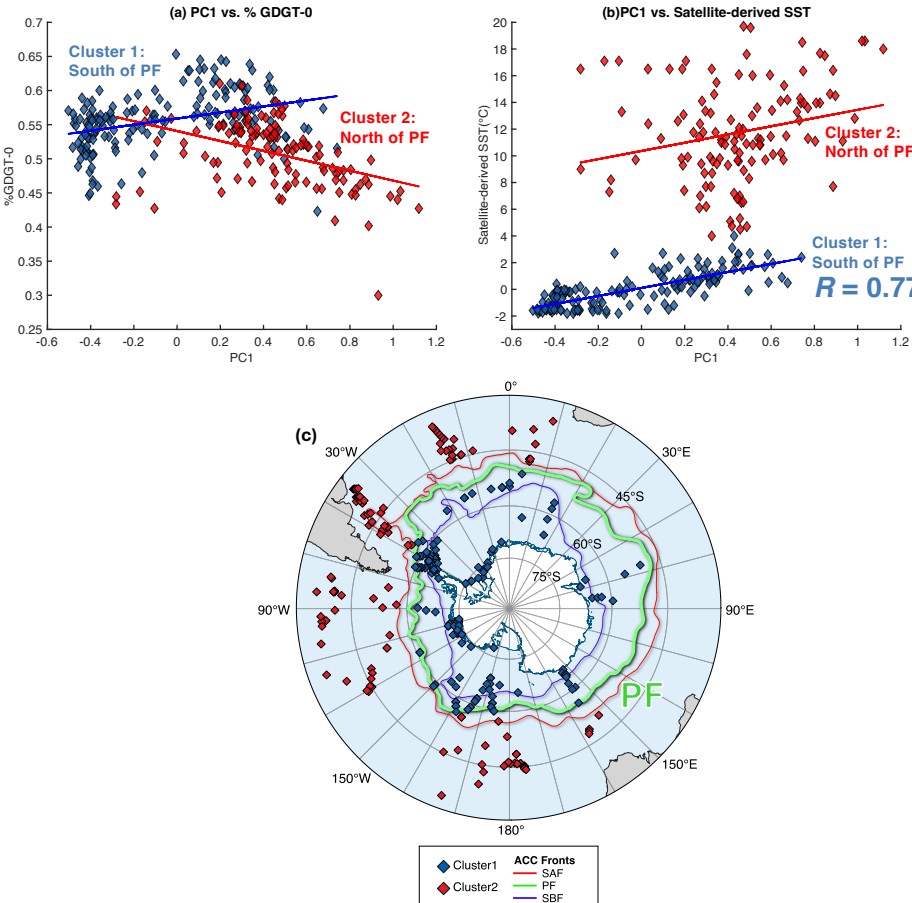

**Figure. 3: Two clusters separated by PF (*n* = 289). (a) Correlation of Southern Ocean PC1 and %GDGT-0; (b) Correlation of Southern Ocean PC1 and satellite-derived in situ SST for the core-top data. (c) Map showing the locations of the Southern Ocean core-top samples south of 35° S. Red diamonds represent the sites north of the PF, while blue diamonds represent the sites south of PF.**

**Table 2: Correlation coefficients (*R*) between principal components (PC1, PC2) and environmental parameters/GDGT indices for the Southern Ocean (SO) (*n* = 289) and Global (*n* = 911) datasets.**

| | Environmental parameters | | |
|---|---|---|---|
| | SST | Salinity | Oxygen |
| PC1 SO (70.3%) | **0.69** | 0.33 | -0.60 |
| PC2 SO (22.1%) | 0.40 | 0.42 | 0.53 |
| PC1 Global (69.3%) | **0.89** | 0.34 | -0.85 |
| PC2 Global (20.9%) | -0.05 | 0.17 | 0.01 |
| | GDGT indices | | |
| | MI | fcren | [2]/[3] | %GDGT-0 |
| PC1 SO (70.3%) | **0.94** | **0.89** | **0.74** | -0.25 |
| PC2 SO (22.1%) | -0.35 | 0.01 | -0.21 | -0.94 |
| PC1 Global (69.3%) | **0.73** | **0.86** | 0.38 | **-0.87** |
| PC2 Global (20.9%) | 0.67 | -0.01 | 0.63 | 0.43 |

Values showing stronger correlations are shown in bold.



### 4.3 Oceanographic controls on isoGDGT variability in the Southern Ocean

In the Southern Ocean, water column structure and associated processes such as vertical mixing and upwelling of CDW vary in the different zones delimited by the oceanic fronts (Carter et al., 2022). To explore how these oceanographic features relate to isoGDGT distributions, PC1 values from the Southern Ocean were compared across different water masses defined by the ACC, including south of the ACC (south of the SBF), centre of the ACC (between the SAF and SBF), and north of the ACC (north of the SAF) (Fig. 4c). A boxplot of PC1 values separated by each frontal zones (Fig. 4a) highlights distinct distributions

for each zone, suggesting significant differences associated with zonal water mass structure. Furthermore, the scatter plot of PC1 vs. isoGDGT-based indices shows significant linear correlations, with data points transitioning from south of the ACC (lower left) to north of the ACC (upper right), with centre of the ACC forming an intermediate cluster (Fig. S3a). These results suggest that the spatial variability in isoGDGT in the Southern Ocean is primarily caused by zonal water mass transitions across the ACC. The moderate correlation between PC1 and temperature in the Southern Ocean supports this interpretation,

as temperature also significantly varies across frontal zones. This is consistent with a previous study which shows a shift in composition of intact polar lipid GDGT in the Scotia Sea across well-defined fronts (Spencer-Jones et al., 2021).

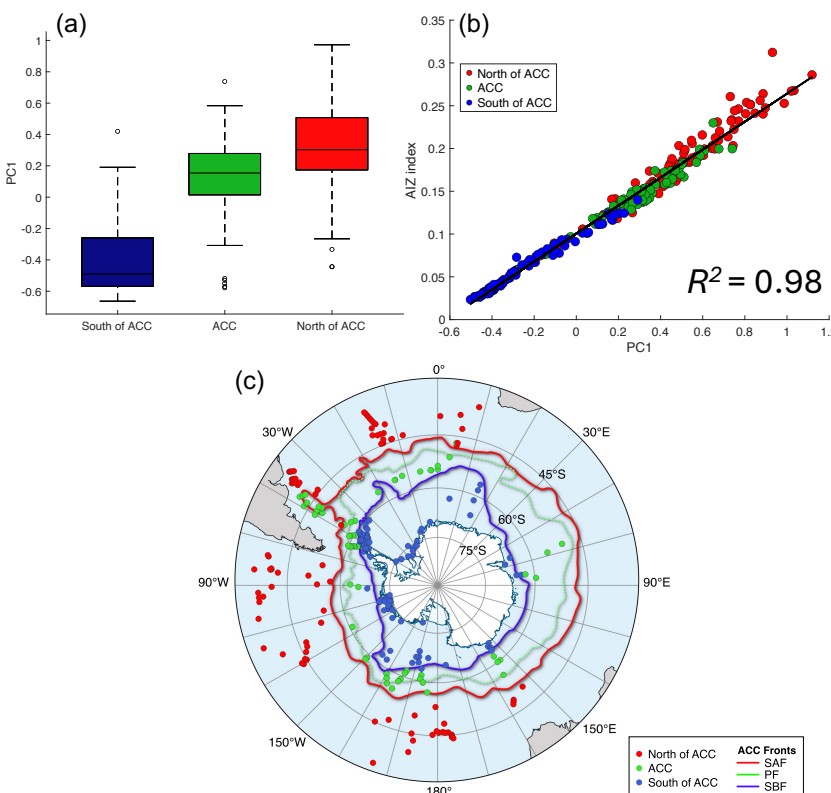

**Figure. 4: PC1 values from the Southern Ocean, separated by frontal zones. (a) Box plot of PC1 in core-top sediments from the Southern Ocean (_n_ = 289), grouped by ACC position. (b) Scatter plot of PC1 versus AIZ index. (c) Map showing the locations of**
**Southern Ocean core-top samples south of 35° S. Blue circles: south of the ACC (south of the SBF); green circles: centre of the ACC (between the SAF and SBF); red circles: north of the ACC (north of the SAF).**




### 4.4 IsoGDGT-based water mass tracer

The PCA of isoGDGT in the Southern Ocean core-top samples showed that PC1 (70.3 % of the variance) primarily reflects temperature and the export depth associated by the zonal water mass properties along the north-south transect across the ACC.

Thus, the isoGDGT which strongly correlate with PC1, could serve as a valid indicator of the zonal water mass in the ACC region. Based on the PCA analysis, we propose a novel isoGDGT-based index as a water mass tracer, composed of GDGT-0, GDGT-1 and GDGT-2, which exhibit significant correlation with PC1. We term this the Antarctic IsoGDGT Zonal (AIZ) index, defined as:

$$AIZ = \frac{[GDGT-1]+[GDGT-2]}{[GDGT-0]+[GDGT-1]+[GDGT-2]} \qquad (7)$$


This new index has a strong linear relationship with PC1 ($R^2 = 0.98$, $n = 289$), with higher values indicating a greater contribution from lower-latitude water masses (Fig. 4b). The index values display a clear gradient, shifting from south of the ACC (lower left) to north of the ACC (upper right), with centre of the ACC clusters in the middle. The AIZ values for each ACC zone are south of the ACC, 0.03–0.08 (±0.03); center of the ACC, 0.13–0.16 (±0.03); and north of the ACC, 0.15–0.21

(±0.04), based on the interquartile range, with error bands representing one standard deviation (Fig. 5). This pattern suggests that the AIZ index effectively tracks the north-south transect of water mass changes across the ACC.

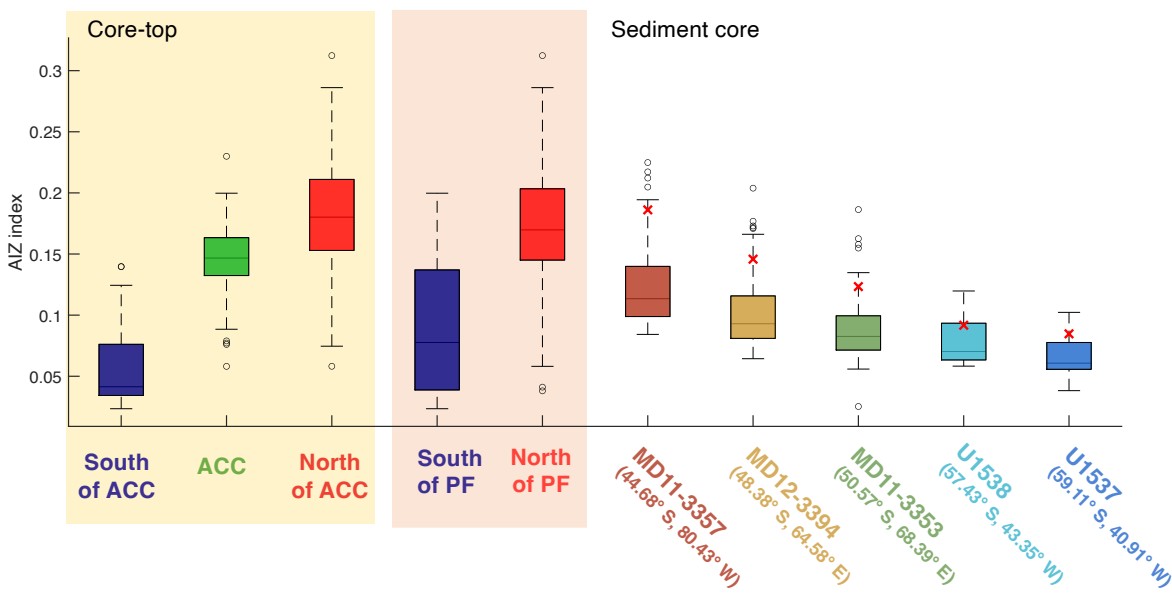

**Figure. 5: Box plot of the AIZ index values in core-top sediments from the Southern Ocean (*n* = 289) and sediment cores (MD11-3357, MD12-3394, MD11-3353, U1538, and U1537). Box elements represent maximum, upper quartile, median, lower quartile, and**

**minimum values. Outliers are shown as open circles; red "x" marks indicate core-top values from each sediment core. The light-yellow background highlights the modern water mass boundary defined by the SBF and SAF (Fig. 4c); the light orange background highlights the boundary defined by the PF (Fig. 3c).**



### 4.5 Regional core-top calibration for AIZ index

The weak correlation between temperature and the conventional isoGDGT-based indices (TEX$_{86}$ and TEX$_{86}^{L}$) in the polar region questions their applicability as reliable temperature proxies in the region (Fietz et al., 2016; Kim et al., 2008). On the other hand, the AIZ index exhibits a relatively strong correlation ($R = 0.77$) with SSTs in the Antarctic Zone, specifically south of the PF (Fig. 3b). This finding suggests that AIZ has a potential as a reliable proxy for reconstructing ocean temperature in the Antarctic Zone.


It has been reported that the seasonality and depth of isoGDGT production is regionally dependent in the Southern Ocean (Church et al., 2003; Park et al., 2019; Sow et al., 2022; Spencer-Jones et al., 2021). To evaluate the potential of AIZ as a temperature proxy, we examined correlations between AIZ values and ocean temperatures at different depths (0–1200 m) across different seasons (summer, autumn, winter, spring, and annual mean) derived from the WOA18 0.25° × 0.25° gridded

product (Boyer et al., 2018). The results show that the correlation varies considerably with depth but is less affected by seasonal changes. Significant correlations ($R^2 = 0.65$–0.82) were observed at depths below 200 m in all seasons, peaking at 400 m depth ($R^2 = 0.77$–0.82) (Table 3, Fig. 6). Notably, the correlation of the logarithmic calibration ($R^2 = 0.67$–0.90) is higher than that of the linear calibration at depths below 200 m (Table 3, Fig. 6). This pattern of improved correlation with non-linear calibrations has been recognised previously for conventional isoGDGT indices in Southern Ocean core-top datasets (Park et

al., 2019). Correlations between AIZ values and both salinity and oxygen concentration at subsurface depths across seasons were also evaluated, but these correlations were much weaker than that between AIZ and ocean temperature ($R^2 = 0.0001$–0.45; Table S1 and S2; Sigs. S9 and S10).

Our results, which show the highest correlation of AIZ index with subsurface temperature, suggest that isoGDGT in the

Southern Ocean are primarily produced at subsurface depths, potentially associated with the archaea inhabiting CDW. CDW is pervasive throughout the water column but centred at approximately 500 m depth south of the PF (Holland et al., 2020), consistent with the 400 m depth where the strongest correlation is observed. Subsurface production of isoGDGT from the Thaumarchaeota in the Southern Ocean has also been suggested by previous work, including 16S rRNA gene (Church et al., 2003; Kalanetra et al., 2009; Sow et al., 2022), living archaea (Spencer-Jones et al., 2021), sediment trap studies (Park et al.,

2019) and sedimentary records (Etourneau et al., 2019; Ho & Laepple, 2016; Kim et al., 2012; Lamping et al., 2021; Liu et al., 2020). Furthermore, it has been reported significant reductions in Thaumarchaeota abundance above 100 m depth in the Southern Ocean (Signori et al., 2014) possibly due to light inhibition (Merbt et al., 2012), which is consistent with the decline in correlation above 200 m depth. These lines of evidence indicate that AIZ index most likely reflects mesopelagic (200–1000 m depth) water temperature. The calibration equations based on annual mean temperature at 400 m depth (subsurface

temperature: Sub-ST) where the strongest correlations (both linear and logarithmic) were obtained, are as follows:

Linear: Sub-ST = 24.17×AIZ-1.45 ($R^2 = 0.81$, $n = 134$, $p < 0.0001$)                                        (8)





Logarithmic: Sub-ST = 2.08 ln(AIZ)+6.11 ($R^2$ = 0.88, $n$ = 134, $p$ < 0.0001)          (9)

It is important to note that AIZ index-based palaeothermometry is only applicable within the Antarctic Zone and is valid for
temperature up to 4 °C (Fig. S6). Therefore, reconstructed temperatures exceeding 4 °C using this calibration represent
extrapolated values. In addition, this calibration does not work well when the sediment core site is significantly affected by
intrusion of water mass from north of the PF, where the temperature-AIZ relationship differs from that of south of the PF.

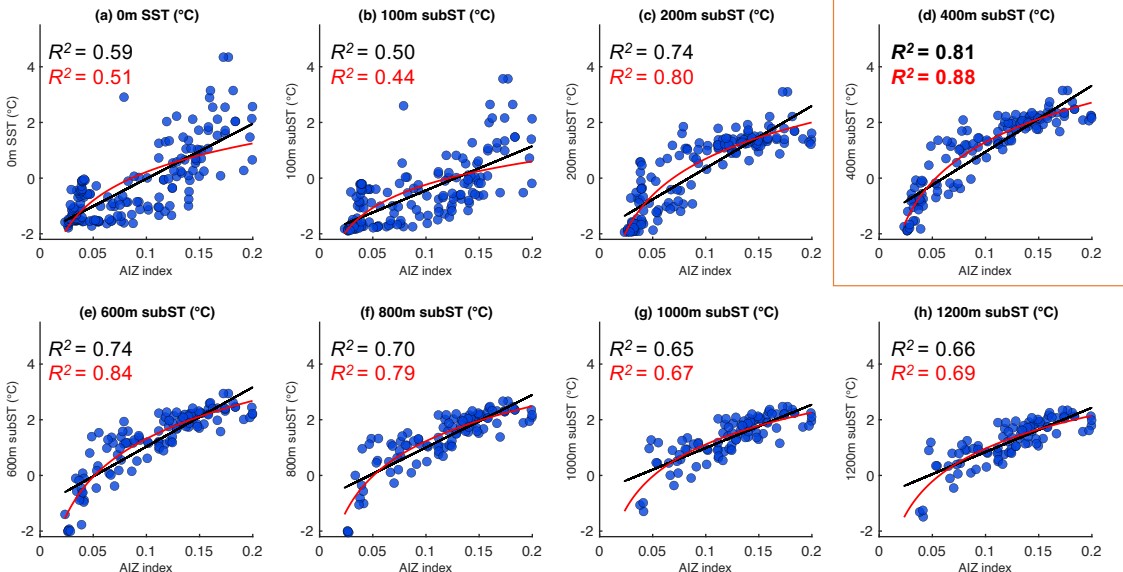

**Figure. 6: Scatter plot of core-top AIZ index versus satellite-derived in situ mean annual ocean temperature at various depths (0–**
**1200 m) south of the PF ($n$ = 96–168). Linear calibration lines and their $R^2$ values are shown in black, and nonlinear calibration**
**curves and their $R^2$ values are shown in red.**

**Table 3: Coefficients of determination ($R^2$) between AIZ values and water temperatures at different depths in different seasons at**
**sites south of PF.**

| Season / Depth | 0 m | 100 m | 200 m | 400m | 600m | 800m | 1000m | 1200m |
|---|---|---|---|---|---|---|---|---|
| | ($n$ = 168) | ($n$ = 158) | ($n$ = 150) | ($n$ = 134) | ($n$ = 124) | ($n$ =109) | ($n$ = 98) | ($n$ = 96) |
| Annual | 0.59 | 0.50 | 0.74 | **0.81** | 0.74 | 0.70 | 0.65 | 0.66 |
| | (0.51) | (0.44) | (0.80) | **(0.88)** | (0.84) | (0.79) | (0.67) | (0.69) |
| Summer (Jan–Mar) | 0.60 | 0.41 | 0.71 | **0.82** | 0.75 | 0.70 | 0.66 | 0.68 |
| | (0.55) | (0.35) | (0.75) | **(0.90)** | (0.85) | (0.80) | (0.69) | (0.70) |
| Autumn (Apr–June) | 0.66 | 0.60 | 0.76 | **0.77** | 0.74 | 0.69 | 0.63 | 0.64 |
| | (0.58) | (0.56) | (0.81) | **(0.85)** | (0.84) | (0.78) | (0.66) | (0.68) |
| Winter (July–Sep) | 0.49 | 0.50 | 0.76 | **0.78** | 0.74 | 0.69 | 0.65 | 0.66 |
| | (0.38) | (0.43) | (0.83) | **(0.86)** | (0.83) | (0.78) | (0.66) | (0.68) |
| Spring (Oct–Dec) | 0.44 | 0.46 | 0.72 | **0.77** | 0.72 | 0.69 | 0.65 | 0.66 |
| | (0.35) | (0.40) | (0.79) | **(0.85)** | (0.82) | (0.78) | (0.67) | (0.69) |

$R^2$ values for the regression lines and logarithmic curves are shown with and without brackets, respectively. The highest $R^2$ values in
every season are shown in bold.





## 4.6 Applying AIZ index to the Southern Ocean sediment cores

### 4.6.1 Evaluation of AIZ index as a water mass tracer around ACC zone

To further evaluate the applicability of AIZ index as a proxy for water mass and temperature in the Southern Ocean, we applied it to the three previously reported sedimentary isoGDGT records collected from the southern Indian Ocean, and the two newly generated records from the Scotia Sea (Fig. 1). Fig. S7 shows a scatter plot of PC1 and AIZ records in the five sediment cores together with that of the Southern Ocean core-top samples. The PC1 and AIZ indices in the five sediment cores are strongly

correlated ($R^2$ = 0.96–0.98), similar to the Southern Ocean core-top dataset. This demonstrates the high potential of AIZ index as a robust tracer for zonal water masses in the ACC zone. Fig. 7a shows sedimentary records of AIZ index at the five sites over the past 160 kyr. Core-top AIZ values in the sediment cores decrease with increasing latitude and fall within the range of AIZ values for each zonal water mass derived from modern core-top data, which are consistent with the modern positions of the oceanic fronts.


The AIZ records from all five sediment cores reflect the glacial-interglacial (G-IG) cycles over the past 160 kyrs, with lower and higher values in glacial and interglacial periods, respectively (Fig. 7a). This variation is coherent with the temperature anomaly (ΔT) in the Dome Fuji ice core (Uemura et al., 2018) (Fig. 7e), suggesting a coupling of Antarctic temperature and ACC dynamics with north-south migration of the ACC position during the G-IG cycle. During the Last Interglacial, the AIZ

values in cores MD11-3357, MD12-3394 and MD11-3353 correspond to modern values north of ACC, while those in U1538 and U1537 fall within the range of centre of the ACC. This suggests that the ACC migrated ~5° southward during the Last Interglacial compared to its present position. On the other hand, during the Last Glacial Maximum AIZ values in all five sites align with values south of the ACC, suggesting a maximum northward shift of ~10° latitude during the Last Glacial Maximum. These results are consistent with previous studies that also indicate the south-north migration of the ACC during G-IG cycles

(Abelmann et al., 2015; Bianchi & Gersonde, 2002; Chadwick et al., 2020; Civel-Mazens et al., 2021; Gersonde et al., 2005). These results highlight that AIZ index can be a powerful proxy for tracing oceanic front migration around the ACC.




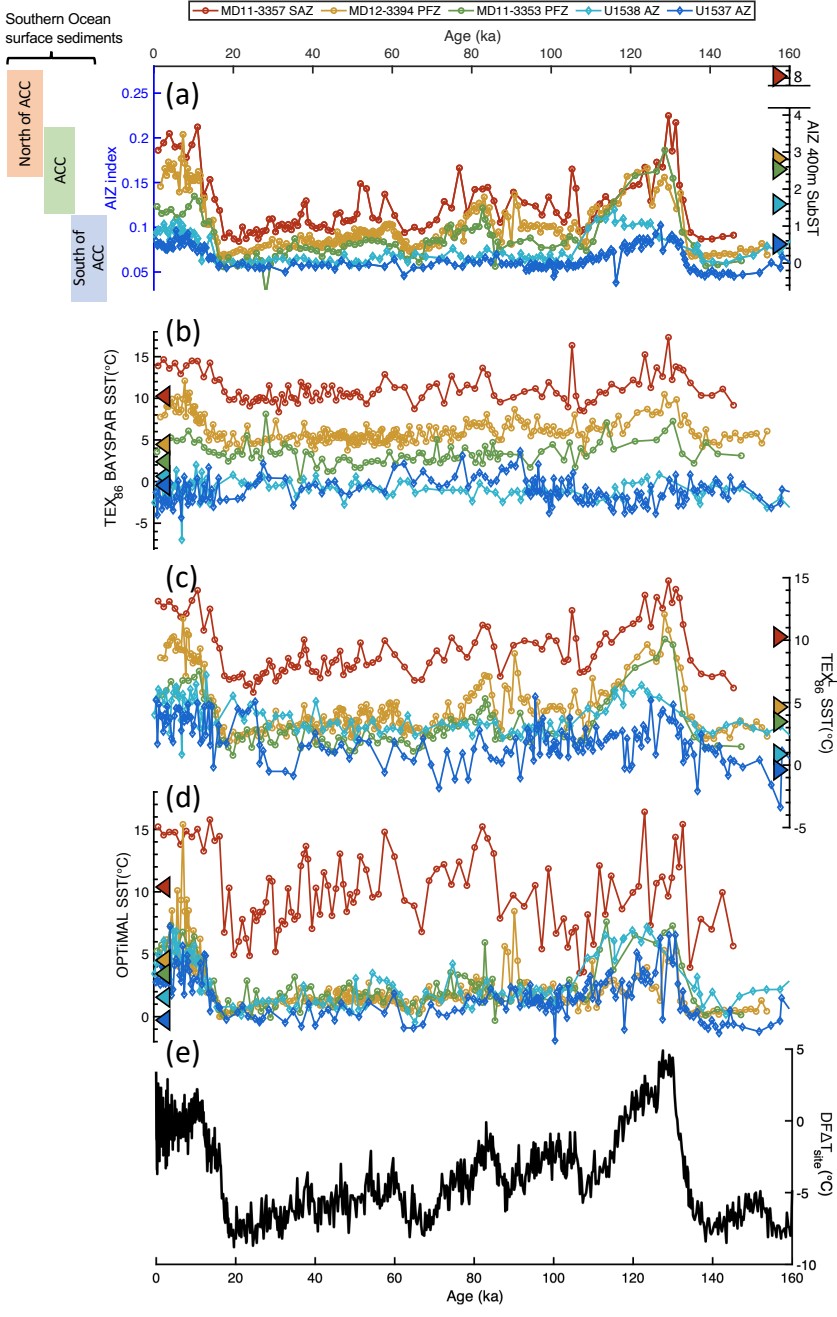

**Figure. 7: IsoGDGT-based SST and subST records in five sediment cores and temperature anomalies in the Dome Fuji ice core (DFΔT$_{site}$) over the past 160 kyr.** (a) AIZ index values and modern reference values for each frontal zone (left axis), and AIZ-based

**subST at 400 m depth (right axis), (b) TEX$_{86}$ BAYSPAR SST, (c) TEX$_{86}^{L}$ SST, (d) OPTiMAL SST, (e) temperature anomalies in Dome Fuji ice core. Symbols: dark red circles (MD11-3357), dark yellow circles (MD12-3394), green circles (MD11-3353), light blue diamonds (U1538), blue diamonds (U1537). Modern SST/subST (400 m) values from WOA18 are shown as triangles. Left panel of Fig. (a) represents the range of the AIZ index values in Southern Ocean surface sediments for each modern oceanic front (south of ACC: 0.04-0.08 (±0.03), ACC: 0.13-0.16 (±0.03), north of ACC: 0.15-0.22 (±0.04), derived from Fig. 5).**



### 4.6.2 Comparing isoGDGT-based temperature records in the Southern Ocean during the last 160 kyr

To assess the potential of the AIZ index as a temperature proxy in the Antarctic Zone, the 400 m depth linear calibration equation (Eq. (8)) was applied to the five sediment cores in the ACC zone. AIZ index-based temperature estimates were then compared with temperatures reconstructed using conventional approaches, including $TEX_{86}$ BAYSPAR, $TEX_{86}^{L}$ and OPTiMAL approaches (Dunkley Jones et al., 2020; Kim et al., 2010; Tierney and Tingley, 2014).

Comparison of temperature records derived from each approach revealed that the performance of these methods varied with latitudes (Fig. 7a–d). All isoGDGT-derived temperature showed clear latitudinal trends, with temperature decreasing towards the high latitude sites, consistent with modern meridional temperature gradient in the region. Comparison of modern and core-top temperature revealed that $TEX_{86}$ BAYSPAR SSTs significantly underestimated modern SSTs by approximately 2 °C at the southern sites (i.e. U1538 and U1537) and overestimated them by approximately 3 °C at the northern sites (i.e. MD11-3357 and MD12-3394) (Table 4 and Fig. 7b). On the other hand, the core-top $TEX_{86}^{L}$-derived SSTs exhibited overestimation at most sites (Fig. 7c), while OPTiMAL SSTs estimates in the core-top samples showed little latitudinal variation for four of the sites and were overestimated at most sites (Fig. 7d). In contrast to these approaches, AIZ-based subsurface temperatures in the core-top samples closely match modern subsurface temperatures except for MD11-3357, which is located north of the PF. The underestimation at the northernmost site arises because the AIZ calibration is designed specifically for the region south of the PF, highlighting the application of AIZ index palaeothermometry is limited to south of the PF.

AIZ- and OPTiMAL-derived SST records at all sites represent a typical G-IG cycle with lower and higher values during glacial and interglacial periods, respectively, consistent with the temperature anomaly (ΔT) record in the Dome Fuji ice core (Uemura et al., 2018). On the other hand, orbital-scale SST variations reconstructed using conventional isoGDGT indices ($TEX_{86}$ and $TEX_{86}^{L}$) differ significantly among sites, and the typical G-IG variation pattern becomes unclear closer to the poles. For instance, the $TEX_{86}$ BAYSPAR SST records show weaker and even anti-phase G-IG variations at the southern sites (MD11-3353, U1538 and U1537). Similarly, $TEX_{86}^{L}$-based SSTs at the southern sites (U1538 and U1537) fail to show distinct G-IG cycles. These results show that the AIZ index seems to provide a more reliable estimate of temperature compared to other approaches in the Antarctic Zone.

The difference in variation patterns observed in G-IG cycles between the traditional isoGDGT-based and AIZ/OPTiMAL-based approaches is likely attributed to the exclusion or inclusion of GDGT-0. Traditional indices exclude GDGT-0, while OPTiMAL and AIZ index, which show a clear G-IG cycle, include GDGT-0 in their calculations. This suggests that GDGT-0 is a key component in reconstructing temperature at southern high latitudes. In fact, the production of GDGT-0 increases with decreasing water temperature. This is explained by the physiological necessity to reduce the number of cyclopentane rings in archaeal membrane lipids to maintain membrane fluidity at low temperatures (Fietz et al., 2020; Gabriel & Lee Gau





Chong, 2000; Schouten et al., 2002). From a physiological perspective, GDGT-0 plays an important role in regulating membrane fluidity in cold environments. Moreover, the correlation between the fractional abundance of GDGT-0 and SST is

also strong in global core-top dataset ($R^2$ = 0.72, $n$ = 894), suggesting that GDGT-0 is intrinsically associated with growth temperature across a wide range of temperature.

In contrast to AIZ index, traditional indices that exclude GDGT-0 (especially TEX$_{86}$), lose their sensitivity to temperature as temperatures decline and as the sites approach the poles (Fig. S8). Consequently, the suitability of these isoGDGT-based

indices for reconstructing temperature at high latitudes is reduced. These results show that AIZ index, which includes GDGT-0, is suitable for estimating temperatures in the Antarctic Zone.

**Table 4: Comparison of isoGDGT indices with reconstructed core-top SST/subST and Glacial-Interglacial (G-IG) cycles.**

| Site | Lat, Lon | | Core-top SST/subST (°C) and G-IG cycles | | | | WOA18 temperature (°C) | |
| --- | --- | --- | --- | --- | --- | --- | --- | --- |
| | | | TEX$_{86}$ | TEX$_{86}^{L}$ | OPTiMAL | AIZ index | 0 m | 400 m |
| MD11-3357 | 44.68°S 80.43°E | Temp | 12.93 | 13.13 | 13.89 | 3.05 | 10.02 | 8.02 |
| | | G-IG | Strong | Strong | Weak | Strong | | |
| MD12-3394 | 48.23°S 64.35°E | Temp | 7.75 | 8.60 | 5.91 | **2.07** | 4.29 | 2.40 |
| | | G-IG | Strong | Strong | Weak | Strong | | |
| MD11-3353 | 50.34°S 68.23°E | Temp | **3.56** | 5.24 | 6.68 | **1.53** | 3.13 | 2.27 |
| | | G-IG | Weak | Strong | Strong | Strong | | |
| U1538 | 57.43°S 43.35°W | Temp | -2.50 | 4.02 | 4.45 | **0.77** | 0.83 | 1.63 |
| | | G-IG | Anti-phase | Weak | Strong | Strong | | |
| U1537 | 59.11°S 40.91°W | Temp | **-1.28** | 5.20 | 4.51 | **0.60** | -0.34 | 0.56 |
| | | G-IG | Anti-phase | Weak | Strong | Strong | | |

The Temp shows the SSTs for TEX$_{86}$, TEX$_{86}^{L}$ and OPTiMAL, and subST at 400m for AIZ index. The G-IG shows the strength of the similarity to the ice core ΔT in orbital-scale variation. AIZ index is calibrated using satellite-derived temperatures at 400 m, whereas other indices are calibrated using satellite-derived temperature at 0m. The core-top values which are within ±1.0 °C of the WOA18 SST/subST are shown in bold.

## 5. Conclusion

1.  Reanalysis of core-top isoGDGT data for the global ocean and the Southern Ocean (south of 35° S) reveals that isoGDGT distributions in the Southern Ocean differ from global patterns.

    2.  IsoGDGT distributions in the Southern Ocean are primarily influenced by temperature and export depth, both shaped by zonal water mass properties across the ACC. The relative abundances of GDGT-0, GDGT-1, and GDGT-2 reflect these water mass-dependent influences. Based on this finding, we propose the AIZ index, a novel isoGDGT-based

420        proxy, which can serve as a water mass tracer within the ACC zone.

    3.  The AIZ index in core-top samples also correlates significantly with ocean temperature at 400 m water depth south of the PF, suggesting its potential for reconstructing past temperatures in the Antarctic Zone.



4. Application of AIZ index to late Pleistocene sediment cores collected from the Southern Ocean confirms its reliability as a tracer of ocean front movement within the ACC zone and temperature proxy in the Antarctic Zone.

5. These findings highlight the potential of isoGDGT in enhancing our understanding of palaeoceanographic conditions in the Southern Ocean, providing a valuable tool for future palaeoceanographic and climate studies.

**Data availability**

All data used in this study are publicly available at https://doi.org/10.5281/zenodo.17060619 (Ishii, 2025). Any further requests for data may be directed to the corresponding author.

**Author contribution**

HI and OS designed the project. HI, OS and MY measured and analysed the data. HI drafted the manuscript with support from OS. All authors discussed the results, commented on the manuscript and approved the final version.

**Competing interests**

The authors declare that they have no conflict of interests.

**Acknowledgements**

Samples and data were provided by the International Ocean Discovery Program (IODP). We thank the captain, crew and IODP staff that made IODP Expedition 382 and subsequent research successful. We offer our special thanks to the lab technicians Kaori Ono and Yuka Nakamura.

**Financial support**

This research is supported by the Japan Society for the Promotion of Science (17H01166, 20H00626, 24K21555) grant awarded to OS funded by the Ministry of Education, Culture, Sports, Science and Technology, Japan.

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
