# Peer review of "New isoprenoid GDGT index as a water mass and temperature proxy in the Southern Ocean"

_EGUsphere, 2025_

## Referee Comment (RC2)

Review: New isoprenoid GDGT index as a water mass and temperature proxy in the Southern Ocean Hana Ishii, Osamu Seki, Masanobu Yamamoto, Bella Duncan

This manuscript presents a new index and calibration for GDGTs based on surface sediments in the Southern Ocean. The authors used PCA biplots to determine the main factors that influence the GDGT distribution. Furthermore, they used the new index to correlate it with water temperatures at different depths and found the best correlation at 400m. In a last chapter, they compare their new calibration with previous calibrations on five sites in the Indian and Atlantic Ocean.

Overall, the article is good, especially the PCA analysis, and I appreciate the publication, even though I think the article needs a bit more work and reorganization.

- 1. The introduction lacks some important information about GDGTs. For example, that the number of rings decreases with decreasing temperatures, which becomes important later in the article. I also think it would be useful to include the current debate about whether the signal is a depth signal or a surface signal, as the type of calibration (SST or subST) has an enormous influence on the results. It should be noted here that this discussion has not been adequately addressed in the SO so far. Since the authors discuss depth in detail in the later part, I would even include this in the scientific question here.
- 2. I also believe that the authors should compare the indices (Figure S2) with subsurface temperatures. It is not surprising that satellite data, which only reflect the top few centimeters of the ocean (and thus the cold freshwater cover), show little correlation with the GDGT indices in the SO, which occur throughout the water column. If possible, I would add a second graph here showing that the already existing indices do also not correlate with subsurface data. In this context, you could refer to the data from Kim et al. (2012), which present a TEX86L subsurface calibration based on WOA09 data. Kim et al. (2012) <a href="https://doi.org/10.1029/2012GL051157">https://doi.org/10.1029/2012GL051157</a>
- 3. In the last chapter when comparing all cores the SSTs derived from satellite data are compared with the subSTs based on the new calibration. Based on the results that GDGTs reflect a subsurface signal there, the comparison is biased from the start. I recommend two articles (Kim et al., 2012 and Hagemann et al., 2023; https://doi.org/10.5194/cp-19-1825-2023), which also present a TEX86L-based subsurface calibration for the SO for a TEX86L based comparison instead.

**Minor comments:**

Line 34: add Reference Brochier-Armanet et al., 2008, who named the phylum of the *Thaumarchae-ota* <a href="https://www.nature.com/articles/nrmicro1852v">https://www.nature.com/articles/nrmicro1852v</a>

Line 38: write isomere instead of regioisomere.

Line 81: Change last part of the sentence and add unit e.g., "In this zone, SSTs drop below 4°C with a salinity around ~34.0 **PSU**."

Line 83: change: "with a salinity of" to "reaching near-freezing temperatures at a continuous salinity of 34.0 **PSU**"

Line 88: I would split the sentence, "but" is irritating -> "...extends from  $^{\sim}1400$  m to  $^{\sim}3500$  m depth. South of the PF, CDW upwells ..." OR "...extends from  $^{\sim}1400$  m to  $^{\sim}3500$  m depth, with upwelling of the CDW south of the PF ..."

Line 90: Just a general comment: the upwelling event might be driven mainly by the Westerlies (haven't checked out the Carter paper yet), but the general driver of the entire ACC are the Westerlies in combination with buoyancy forcing (Rintoul, 2018 <a href="https://doi.org/10.1038/s41586-018-0182-3">https://doi.org/10.1038/s41586-018-0182-3</a>), maybe you want to mention it.

Line 96: The "are also indicated" is unnecessary. I would simply delete them. Anyway, if it is possible, I would add the long version like "Subtropical Front" to the legend, behind the shortcut STF and delete them fully of the description. And then I would write instead the Figure capture as: "Bathymetric map with sediment core location and oceanic fronts analyzed in this study."

Line 374: Why did you compared it with Kim 2010, which was a surface calibration and not with the subsurface calibration after you figured out that it is probably a subsurface signal?

Line 388: OPTiMAL derived SSTs of MD11-3351 look to me very noisy, especially during the glacial period, with temperatures compatible to LIG. This is not a typically G-IG pattern. I would add the wording "almost every site except MD11-3357, which shows temperatures during the glacial comparable to the Interglacials."

**Table 4:**

- Generally, if you do a sediment surface comparison, it is better to use WOA05 or WOA09 since global warming has less impact on the surface temperatures there.
- The placeholder characters are irritating.
- Satellite-derived temperatures at 400m? Is that possible? As far as I know it is only the surface. <a href="https://podaac.ipl.nasa.gov/SeaSurfaceTemperature">https://podaac.ipl.nasa.gov/SeaSurfaceTemperature</a>

If you have any questions, feel free to contact me.

Julia R. Hagemann (Julia. Hagemann@marine.rutgers.edu)

---

## Author Comment (AC2)

Reply: We thank the reviewer for their thorough evaluation and positive assessment of our work. Below we indicate with each comment how we intend to adjust the manuscript. Below, we list each of the comments provided by the reviewer in bold black text and our response to them in plain blue text.

**This manuscript presents a new index and calibration for GDGTs based on surface sediments in the Southern Ocean. The authors used PCA biplots to determine the main factors that influence the GDGT distribution. Furthermore, they used the new index to correlate it with water temperatures at different depths and found the best correlation at 400m. In a last chapter, they compare their new calibration with previous calibrations on five sites in the Indian and Atlantic Ocean.**

**Overall, the article is good, especially the PCA analysis, and I appreciate the publication, even though I think the article needs a bit more work and reorganization.**

1. **The introduction lacks some important information about GDGTs. For example, that the number of rings decreases with decreasing temperatures, which becomes important later in the article. I also think it would be useful to include the current debate about whether the signal is a depth signal or a surface signal, as the type of calibration (SST or subST) has an enormous influence on the results. It should be noted here that this discussion has not been adequately addressed in the SO so far. Since the authors discuss depth in detail in the later part, I would even include this in the scientific question here.**

   Reply: We appreciate this helpful feedback on the introduction. We agree that additional background information would strengthen the manuscript. We will expand the introduction to include:

   (1) a more detailed explanation of the number of rings/structures in GDGTs and their significance by adding the following sentences.

   "The degree of cyclisation of isoGDGT is strongly correlated with ocean temperature in global core-top datasets (Schouten et al., 2002), attributed to adaptive changes in archaeal membrane ring structures that maintain optimal fluidity under varying ambient temperatures (Fietz et al., 2020; Gabriel & Chong, 2000; Schouten et al., 2002). In low temperature environments, archaea reduce the number of cyclopentane rings in their membrane structures, to prevent membrane rigidity. Based on this relationship, Schouten et al. (2002) proposed the first isoGDGT index ($TEX_{86}$)."

   (2) Adding a discussion of the current debate regarding the most appropriate water mass depth for calibration (SST vs. subST), especially in the SO. We will add the following paragraph into introduction.

   "There is an ongoing debate about the depth of production and export of sedimentary GDGTs. In the Antarctic Zone, GDGTs are thought to predominantly reflect subsurface rather than surface ocean temperatures (Fietz et al., 2016; Hagemann et al., 2023; Ho and Laepple, 2016; Jaeshke et al., 2017; Kim et al., 2012; Lamping et al., 2021; Park et al., 2019). This is supported by observations of elevated archaeal abundances in Circumpolar Deep Water (Alonso-Sáez et al., 2011; Church et al., 2003; Kalanetra et al., 2009; Sow et al., 2022; Spencer-Jones, 2021). These genomic studies have also revealed differences in archaeal communities and archaeal diversity across water masses, reflecting the influence of oceanographic features (Kolody et al., 2025; Raes et al., 2018). Additionally, the seasonality of isoGDGT production (Chandler

& Langebroek, 2021; Church et al., 2003; Park et al., 2019) and polar-related biases such as seasonal change in sea ice (Xu et al., 2020) have been thought to contribute to GDGT variability.

2. **I also believe that the authors should compare the indices (Figure S2) with subsurface temperatures. It is not surprising that satellite data, which only reflect the top few centimeters of the ocean (and thus the cold freshwater cover), show little correlation with the GDGT indices in the SO, which occur throughout the water column. If possible, I would add a second graph here showing that the already existing indices do also not correlate with subsurface data. In this context, you could refer to the data from Kim et al. (2012), which present a TEX86L subsurface calibration based on WOA09 data. Kim et al. (2012) https://doi.org/10.1029/2012GL051157**
   Reply: We thank the reviewer for this comment. We will analyse the relationship between traditional indices and subsurface temperatures and incorporate this into Figure S2.

3. **In the last chapter - when comparing all cores - the SSTs derived from satellite data are compared with the subSTs based on the new calibration. Based on the results that GDGTs reflect a subsurface signal there, the comparison is biased from the start. I recommend two articles (Kim et al., 2012 and Hagemann et al., 2023; https://doi.org/10.5194/cp-19-1825-2023), which also present a TEX86L-based subsurface calibration for the SO for a TEX86L based comparison instead.**
   Reply: We thank the reviewer for this suggestion. We agree that comparing SST-calibrated indices with subsurface temperatures may not provide a fair assessment. We recalculated $TEX_{86}^{L}$ temperatures using the Kim et al. (2012) subsurface calibration (0-200 m) for MD11-3357 (north of SAF), and the Hagemann et al., 2023 calibration for the four sites south of SAF. The recalculated temperature values are compared with WOA18 temperatures at subsurface depths (e.g., 0-200 m). As a result, these recalculated values, especially using Hagemann et al. (2023) calibration, improved the reconstructed temperature range. However, our conclusions regarding the AIZ index and its calibration remain unchanged, as it shows the best performance in terms of temperature range and representation of glacial-interglacial cycles. This provides a more equitable comparison between traditional indices and our new index, with each now applied at their intended depth range.

**Minor Comments:**

**Line 34: Add Reference Brochier-Armanet et al., 2008, who named the phylum of the *Thaumarchaeota* https://www.nature.com/articles/nrmicro1852v**
Reply: We will add the reference to the revised manuscript.

**Line 38: Write isomere instead of regioisomere.**
Reply: We will replace te term "regioisomer" with "stereoisomer."

**Line 81: Change last part of the sentence and add unit e.g., "In this zone, SSTs drop below 4°C with a salinity around ~34.0 PSU."**
Reply: We will change this accordingly.

**Line 83: Change: "with a salinity of" to "reaching near-freezing temperatures at a continuous salinity of 34.0 PSU"**
Reply: We will change this accordingly.

**Line 88: I would split the sentence, "but" is irritating → "...extends from ~1400 m to >3500 m depth. South of the PF, CDW upwells ..." OR "...extends from ~1400 m to >3500 m depth, with upwelling of the CDW south of the PF ..."**
Reply: We will change this according to the first suggestion.

**Line 90: Just a general comment: the upwelling event might be driven mainly by the Westerlies (haven't checked out the Carter paper yet), but the general driver of the entire ACC are the Westerlies in combination with buoyancy forcing (Rintoul, 2018 https://doi.org/10.1038/s41586-018-0182-3), maybe you want to mention it.**
Reply: We agree and will add a description of ACC circulation drivers (westerly winds and buoyancy forcing) to the oceanographic settings section, following Rintoul (2018).

**Line 96: The "are also indicated" is unnecessary. I would simply delete them. Anyway, if it is possible, I would add the long version like "Subtropical Front" to the legend, behind the shortcut STF and delete them fully of the description. And then I would write instead the Figure capture as: "Bathymetric map with sediment core location and oceanic fronts analyzed in this study."**
Reply: We will change both the text and the figure (legend) accordingly.

**Line 374: Why did you compared it with Kim 2010, which was a surface calibration and not with the subsurface calibration after you figured out that it is probably a subsurface signal?**
Reply: As mentioned above, we recalculated $TEX_{86}^L$ temperatures following Hagemann et al. (2023) recommendations: applying the Kim et al. (2012) subsurface calibration (0-200 m) for MD11-3357 north of the SAF, and the Hagemann et al. (2023) calibration for four sites south of the SAF.

**Line 388: OPTiMAL derived SSTs of MD11-3351 look to me very noisy, especially during the glacial period, with temperatures compatible to LIG. This is not a typically G-IG pattern. I would add the wording "almost every site except MD11-3357, which shows temperatures during the glacial comparable to the Interglacials."**
Reply: We will change this accordingly.

**Table 4**
- **Generally, if you do a sediment surface comparison, it is better to use WOA05 or WOA09 since global warming has less impact on the surface temperatures there.**
  Reply: We appreciate this comment and understand the concern. The reason we chose WOA18 is that it has higher resolution grid (0.25 x 0.25 degrees, compared to WOA09 (1 x1 degrees)). We have also checked the differences between WOA18 and WOA09 (using a 1.0x1.0 degrees grid for both), and found that the temperature differences at the study sites range from - 0.3 ~ +0.4 °C, confirming that these differences do not change our conclusion. Thus, we will continue to use WOA18 (0.25 x 0.25 degrees).

- **The placeholder characters are irritating.**
  Reply: We will delete the placeholder characters.

- **Satellite-derived temperatures at 400m? Is that possible? As far as I know it is only the surface. https://podaac.jpl.nasa.gov/SeaSurfaceTemperature**
  Reply: We appreciate this clarification. The temperature data used in this study are derived from in situ observations compiled in the World Ocean Atlas (WOA18), not satellite-derived sources. We will revise the terminology to 'WOA-derived temperatures' to ensure clarity.

*Additional references (not found in our original submission) referred to by the reviewer and in our response – these will be incorporated into our final revised submission:*

Alonso-Sáez, L., Andersson, A., Heinrich, F., and Bertilsson, S.: High archaeal diversity in Antarctic circumpolar deep waters: Biogeography of Archaea in Antarctic waters, Env. Microbiol. Rep., 3, 689–697, https://doi.org/10.1111/j.1758-2229.2011.00282.x, 2011.

Brochier-Armanet, C., Boussau, B., Gribaldo, S., and Forterre, P.: Mesophilic Crenarchaeota: proposal for a third archaeal phylum, the Thaumarchaeota, Nat. Rev. Microbiol., 6, 245–252, https://doi.org/10.1038/nrmicro1852, 2008.

Hagemann, J. R., Lembke-Jene, L., Lamy, F., Vorrath, M.-E., Kaiser, J., Müller, J., Arz, H. W., Hefter, J., Jaeschke, A., Ruggieri, N., and Tiedemann, R.: Upper-ocean temperature characteristics in the subantarctic southeastern Pacific based on biomarker reconstructions, Clim. Past, 19, 1825–1845, https://doi.org/10.5194/cp-19-1825-2023, 2023.

Kolody, B. C., Sachdeva, R., Zheng, H., Füssy, Z., Tsang, E., Sonnerup, R. E., Purkey, S. G., Allen, E. E., Banfield, J. F., and Allen, A. E.: Overturning circulation structures the microbial functional seascape of the South Pacific, Science, 2025.

Raes, E. J., Bodrossy, L., Van De Kamp, J., Bissett, A., Ostrowski, M., Brown, M. V., Sow, S. L. S., Sloyan, B., and Waite, A. M.: Oceanographic boundaries constrain microbial diversity gradients in the South Pacific Ocean, Proc. Natl. Acad. Sci. U.S.A., 115, https://doi.org/10.1073/pnas.1719335115, 2018.

Rintoul, S. R.: The global influence of localized dynamics in the Southern Ocean, Nature, 558, 209–218, https://doi.org/10.1038/s41586-018-0182-3, 2018.

Xu, Y., Wu, W., Xiao, W., Ge, H., Wei, Y., Yin, X., Yao, H., Lipp, J. S., Pan, B., and Hinrichs, K.: Intact Ether Lipids in Trench Sediments Related to Archaeal Community and Environmental Conditions in the Deepest Ocean, JGR Biogeosciences, 125, e2019JG005431, https://doi.org/10.1029/2019JG005431, 2020.

---

## Author Comment (AC3)

**Dear Reviewer #1:**
We thank the reviewer for their attention to detail and considerate, helpful comments that will significantly contribute to improving the manuscript. Below, we list each of the comments provided by the reviewer in bold black text and our response to them in plain blue text.

**Ishii and co-authors have compiled isoGDGT data for the Southern Ocean to evaluate their relationship with temperature and water masses. The Southern Ocean is region sensitive to climate change, but reliable temperature reconstructions from this area are scares due to poor preservation of carbonates and deviating temperature sensitivity of lipid biomarker proxies compared to global trends.**

**They authors find that isoGDGTs in surficial sediments are different in sediments north and south of the Polar Front, and also have a different relationship with temperature. They subsequently propose an index that can be used to reconstruct the position of the Polar Front, and possibly reconstruct paleotemperatures.**

**The development of proxies to reliably reconstruct past temperatures in the Southern Ocean is very important. For that reason, I would say that this manuscript is worth publishing in Climate of the Past. However, the current version lacks crucial background information and context at several instances. Also, GDGTs are often written in singular form -also when the are referred to as plural-, which creates unnecessary confusion and ambiguity in the text. Finally, the text contains 'the' at places where it is not necessary, and misses this word at places where it is necessary. Please carefully reread the text before resubmission. Besides from these textual comments, my main points are:**
Reply: We will carefully revise the manuscript to use plural forms (isoGDGTs) and correct all grammatical mistakes, including article usage errors. A native English-speaking colleague will review the final version before resubmission.

**I am missing a mechanism supporting the definition for the proposed AIZ index. From the very \*last\* section of the discussion, it finally appears that GDGT-0 is a crucial compound in driving this index, but this is far too late in the manuscript. Please (at least try to) explain why GDGT-0 occurs more/less in certain water masses and state its relationship with temperature earlier on. This should also be mentioned in the abstract. Important here is to mention that GDGT-0 is deliberately not included in the TEX86 because it is produced by such a wide range of archaea (Schouten et al., 2002 EPSL, Schouten et al., 2013 Org Geochem). We as community should move on from merely reporting statistical relationships to truly understanding the mechanisms that drive TEX86 and GDGT distributions in the environment (and cultures) in general.**
Reply: We appreciate this feedback. Below we address each question mentioned above.

**I am missing a mechanism supporting the definition for the proposed AIZ index.**
Reply: We hypothesize that the potential primary control on the AIZ is change in archaeal community between the water masses. As the reviewer pointed out, this was not mentioned in the first draft, and we will add the following discussion to the revised version.

"Genomic studies in the South Pacific have identified that oceanographic features, such as wind-driven circulation at the surface, are the primary drivers of prokaryotic richness and community diversity patterns (Kolody et al, 2025; Raes et al., 2018). North of the PF, archaeal richness increases northward and peaks at the STF border, which also acts as an ecological boundary (Raes et al., 2018). South of the PF, the diversity of the archaeal community is low, possibly due to the extremely cold and harsh conditions that may have acted as a habitat bottleneck (Alonso-Sáez et al., 2011). Genomic analysis in waters south of the PF shows "*Candidatus Nitrosopumilus maritimus*" dominates the *Nitrososphaera*

phylum (Hernández et al., 2015; Kim et al., 2014; Sow et al., 2021). Moreover, Spencer-Jones et al. (2021) conducted a principal component analysis of intact polar lipids (IPL)-GDGT compositions in the Amundsen and Scotia Seas, along with previously published clusters of *Nitrososphaera* (Bale et al., 2019), and found clustering within the *Nitrosopumilales* group in both regions due to the high relative abundances of GDGT-0. We suspect that the AIZ index captures these changes in archaeal community composition, specifically reflecting the dominance of cold-adapted *Nitrosopumilales* south of the PF in contrast to more diverse communities northward. Because these community shifts are tied to water mass boundaries, the AIZ index can serve as a good indicator for reconstructing past PF movements."

**From the very \*last\* section of the discussion, it finally appears that GDGT-0 is a crucial compound in driving this index, but this is far too late in the manuscript.**
Reply: We thank the reviewer for this feedback. We will bring the discussion about GDGT-0 to section 4.4 which introduces that the AIZ-index includes GDGT-0.

**Please (at least try to) explain why GDGT-0 occurs more/less in certain water masses and state its relationship with temperature earlier on. This should also be mentioned in the abstract.**
Reply: We appreciate this critical feedback. As described above, we suggest that the importance of GDGT-0 to the AIZ index is due to archaeal community differences between water masses north and south of the PF. The region south of the PF has lower archaeal diversity dominated by the *Nitrosopumilales* group, which is characterized by high relative abundances of GDGT-0 (Santoro et al., 2015; Spencer-Jones et al., 2021). Thus, we presume that this is an important reason for the higher abundance of GDGT-0 (as well as the distinct %GDGT-0 correlation) south of the PF. We will mention this potential mechanism in the abstract and discussion, emphasizing how water mass-driven changes in archaeal communities may enable the AIZ index to serve as a proxy for the PF position. As for the relationship with temperature, please see our response to the following comment.

**Important here is to mention that GDGT-0 is deliberately not included in the TEX86 because it is produced by such a wide range of archaea (Schouten et al., 2002 EPSL, Schouten et al., 2013 Org Geochem). We as community should move on from merely reporting statistical relationships to truly understanding the mechanisms that drive TEX86 and GDGT distributions in the environment (and cultures) in general.**
Reply: We thank the reviewers for this comment, and agree that it is important for the community to move towards a more mechanistic understanding of the proxy. We will mention the reason why GDGT-0 is not included in the $TEX_{86}$, but also discuss why it appears important in the Southern Ocean by adding explanation below, acknowledging that this is from a statistical rather than mechanistic perspective (see response to L40).

"GDGT-0 has traditionally been excluded from temperature indices. This exclusion stems from concerns about both its overpowering influence on index calculations and potential alternative sources besides *Nitrososphaera*. The fractional abundances of GDGT-0 are generally much higher than those of other GDGTs, which would otherwise overpower the index calculations (Schouten et al., 2002; Kim et al., 2010). Thus, one of the reasons for this exclusion is based on mathematical rather than ecological considerations. However, previous studies suggest that GDGT-0 does hold temperature information (Dunkley-Jones et al., 2020; Kim et al., 2010; Zhao et al., 2025). The fractional abundance of GDGT-0 shows a strong correlation with SST in global core-top datasets ($R^2 = 0.72$, $n = 894$), indicating that GDGT-0 is intrinsically associated with growth temperature across a wide temperature range. In fact, the production of GDGT-0 increases with decreasing water temperature (Kim et al., 2010). This is explained by the physiological necessity to reduce the number of cyclopentane rings in the archaeal membrane lipids to maintain membrane fluidity at low temperatures (Fietz et al., 2020; Gabriel & Lee GauChong, 2000; Schouten et al., 2002). Moreover, Dunkley-Jones et al., (2020) found GDGT-0, along with GDGT-3 to be, the most informative GDGTs for predicting temperature using Gaussian process

regression. Therefore, the inclusion of GDGT-0 becomes particularly important at high latitudes, where the relative abundance of GDGT-3 is substantially reduced. Another reason that GDGT-0 has traditionally been excluded from $TEX_{86}$ is the potential influence of factors other than temperature on its abundance in sediments. There are alternative sources of GDGT-0, such as methanogenic and methanotrophic archaea living in anoxic sedimentary environments (Pancost et al., 2001; Zhang et al., 2011). However, methanotrophic archaea likely produce not only GDGT-0 but also substantial amounts of GDGTs-1, -2, and -3 (Pancost et al., 2001; Schouten et al., 2013). We acknowledge the potential impact of non-*Nitrososphaera* sources when using GDGT-0. Nevertheless, by combining screening tests (%GDGT-0 and MI), these non-thermal influences can be minimized. We note that the Southern Ocean dataset we used here to establish the AIZ index was all screened by using %GDGT-0 and MI."

**The AIZ index is used both to recognise water masses and to reconstruct temperature. However, there is no discussion on how these two different aspects can be disentangled. What is the primary control on the AIZ? And how does temperature (and all other environmental parameters mentioned in the text) vary between water masses? In short, the characteristics of the different water masses need to be better described. In the introduction and the Oceanographic setting, and should be used as context for the interpretation of surficial sediment data and downcore proxy results.**
Reply: We appreciate this feedback. Below we address each question mentioned above.

**What is the primary control on the AIZ? And how does temperature (and all other environmental parameters mentioned in the text) vary between water masses?**
Reply: As we addressed in response to the previous comment, we propose that the AIZ index responds to a change archaeal community delineated by water mass change across the PF. When values are 0.14 (± 0.03 SD) or higher, this indicates depositions north of the PF and the AIZ cannot be used to reconstruct temperatures. However, when values are 0.14 or lower, this indicates deposition south of the PF where the archaeal community is dominated by *Nitrosopumilales* group. In this situation, the index varies with temperature and a robust temperature calibration can be established. We will add the following explanation to support the usage of the AIZ index to reconstruct the temperature south of the PF:

"We propose that the AIZ index responds to a change archaeal community delineated by water mass change across the PF. We suggest a threshold value of 0.14 (± 0.03 SD), based on the present PF position. Values above this indicate deposition in waters north of the PF, and cannot be used to calibrate to temperature. Values below 0.14 (± 0.03 SD) indicate deposition south of the PF, and variations in the AIZ index reflect temperature changes. South of the PF, the source of GDGTs appears to be predominantly the *Nitrosopumilales* group (Alonso-Sáez et al., 2011; Hernández et al., 2015; Kim et al., 2014; Sow et al., 2021; Spencer-Jones et al., 2021) and consequently, GDGT compositions show a strong correlation with temperature."

**In short, the characteristics of the different water masses need to be better described. In the introduction and the Oceanographic setting, and should be used as context for the interpretation of surficial sediment data and downcore proxy results.**
Reply: We will expand Section 2 (Oceanographic settings) to provide more comprehensive water mass characterization that better supports our isoGDGT interpretations. The revised section will include detailed subsurface water mass structure and their relationship to nutrient and primary productivity distributions that influence isoGDGT production and export depth. These expanded water mass descriptions will be directly referenced throughout our Results and Discussion sections, particularly in interpreting elevated [2]/[3] ratios (Section 4.2), PC1 variations across ACC zones (Section 4.3), the 400 m depth calibration reflecting CDW influence (Section 4.5), and downcore AIZ variations as indicators of frontal migration (Section 4.6).

**The introduction is a bit sloppy and misses some crucial information and details on the development and use of the TEX86 as proxy for SST.**
Reply: We appreciate this helpful feedback on the introduction. Below we indicate with each comment how we intend to adjust and expand the introduction.

**L36: there are many more than 6 different isoGDGTs (isoGDGTs exist with up to 8 cyclisations, with additional branches, with hydroxy groups, with a c-c bond, etc etc). Please be correct in your statements.**
Reply: We will change this accordingly.

**L37: cren only has 1 cyclohexane moiety.**
Reply: We will change this accordingly.

**L38: cren' is a stereoisomer, not a regioisomer (see Liu et al 2018 Org Geochem and Sinninghe Damsté et al 2018 Org. Geochem).**
Reply: We will change this accordingly.

**L38: specify that it is the number of cyclisations that is related to temperature, not the isoGDGT distribution in general.**
Reply: We will change this accordingly.

**L40: explain why there was a need for studies to further improve the proxy. What were the problems?**
Reply: We will add the below explanation about concerns with applying $TEX_{86}$ and $TEX_{86}^L$ in (sub)-polar environment:

L39-40: "Based on this relationship, Schouten et al. (2002) proposed the first isoGDGT index ($TEX_{86}$). However, subsequent research on $TEX_{86}$ palaeothermometry has raised concerns about applying the $TEX_{86}$ index in cold regions. This stems from its insensitivity to temperature changes in cold environments and large variability in core-top data (Kim et al., 2008; 2010). Various approaches have tried to better constrain GDGT-based temperatures at high latitudes, for example by using additional high latitude core-top data, spatially varying calibrations, or alternative indices such as $TEX_{86}^L$ (Shevenell et al., 2011, Kim et al., 2010, 2012, Tierney and Tingley et al., 2014, 2015). However, in the case of $TEX_{86}^L$, the lack of a physiological basis for the index has seen it not recommended for continued use (Bijl et al., 2025), and while increased core-top data has created a better spatial network of information, the significant scatter and insensitivity to temperature in $TEX_{86}$ values throughout the region has not been better constrained. In the absence as yet of a more mechanistic understanding of how and why GDGT-temperature relationships vary through different temperature zones and *Nitrososphaera* communities, we take a statistical approach to determine an alternative index for a circum-Antarctic GDGT-temperature relationship."

**L46: $TEX_{86}^L$ is not generally used by the community and it is not recommended for use. The Southern Ocean seems to be an exception as unique region where some OK results have been obtained. Given the prominent role of $TEX_{86}^L$ in this manuscript, some background information is warranted. See community discussion (section 7.2) in Bijl et al., in press, Biogeosciences (https://doi.org/10.5194/egusphere-2025-1467).**
Reply: We thank the reviewer for this important point and will add more background on $TEX_{86}^L$ in the introduction, including discussion of its limitations (see response above). We agree that $TEX_{86}^L$ is problematic, which is part of our motivation for developing the AIZ index. Here, we use $TEX_{86}^L$ for comparison to show how our new index performs relative to commonly applied methods in the Southern Ocean.

**L50 states that polar-specific factors should be taken into account, but it is not explained what their influence on TEX86 is.**
Reply: We appreciate this feedback. We stated that polar-specific factors result in a weak $TEX_{86}$-SST correlation, particularly below 5°C (lines 45-47, Fig. S2), and identified these factors in lines 48-50. To improve clarity, we will add a sentence explicitly explaining that we infer that non-thermal influences (such as community composition, habitat depth, and seasonality) result in scatter in high latitude $TEX_{86}$ values and reduced temperature sensitivity in polar waters. The following sentence will be added after L50:

"We infer that large scatter in polar $TEX_{86}$ values could be influenced by environmental conditions other than temperature, or polar archaeal-community specific GDGT-temperature relationships, thereby weakening the temperature signal and reducing $TEX_{86}$ sensitivity to SST variations in cold polar waters (<5 °C; Fig. S2). This scatter could also be influenced by a lack of constraint on core-top ages, which could result in older GDGTs being mixed into modern material (Bijl et al., 2025). "

**L53 mentions hydroxy GDGTs as promising addition to the TEX86 in polar regions, but leaves it at that. Explain the OH-TEX86 (Varma et al., 2024 GCA) and how it extends the linear response of GDGTs to temperatures <15C. This is one of the major advancements of the field of the past years and warrants more discussion and attention. I would even recommend including OH-GDGTs in the analyses here.**
Reply: We thank the reviewer for this suggestion. We agree that OH-GDGTs are an important advancement for polar palaeothermometry and we will expand our introduction to discuss OH-$TEX_{86}$ and its applicability to temperature <15 °C (Varma et al., 2024).

Our study focuses on developing new applications for isoGDGTs, for which substantial datasets are already available in both core-top and downcore datasets in the Southern Ocean. To connect our work with OH-GDGT approaches, we will add analysis of the relationship between OH-GDGTs and the AIZ index using the Lamping et al. (2021) dataset ($n$ = 66) where both compounds were measured. For these data, AIZ index and OH-TEX86 shows a strong correlation with **$R^2$ = 0.94**. Other OH-related indices also show good correlations listed as follows: %OH ($R^2$=0.64), RI-OH ($R^2$=0.79), RI-OH' ($R^2$=0.64), and $OH^c$ ($R^2$=0.94). We also agree and recommend the measurement of OH-GDGTs for future work, but suggest that the AIZ index can be useful for the many instances where isoGDGT data is already published or measured, but OH-GDGTs are not available.

**L62: the Polar Front appears out of nowhere here. The different water masses and their characteristics need to be explained earlier on, possibly in relation with the deviating temperature-relationship of isoGDGTs compared to that in global oceans.**
Reply: We will add information of the oceanic front in the introduction and refine section 2 (oceanographic settings). Also, we will emphasize how south of the PF is essential for understanding the interaction between the ocean and ice sheets.

**I refered to the community-led paper sharing the best practices on the analysis and interpretation of isoGDGTs in marine sediments (Bijl et al., in press, see link above). It would make sense to follow those best practices here, too. For example in assessing outliers or for selecting the most appropriate calibration.**
Reply: We thank reviewer for referring the paper. We will add this paper into our discussion. We note that the Southern Ocean and global dataset we used here to establish the AIZ index was all screened by using %GDGT-0 and MI.

**Section 4.2 mentions export depth, but does not explain the principle, or what determines/influences export depth. Please add more context and explanation here, and make clear how export depth relates to the depth of maximum isoGDGT production, as well as whether these depths are expected to vary between water masses (and if so, how, and why).**

Reply: We thank the reviewer for this suggestion. We will amend section 4.2 with more explanation about GDGT [2]/[3] ratio and how it relates to water masses changes, and we will add the following discussion to the revised version.

"One notable distinction between global and Southern Ocean datasets lies in the relationship between PC1 and [2]/[3]. The [2]/[3] ratio is used to differentiate between contribution from 'shallow' (0-200 m depth) and 'deep' (> ~1000 m depth) clades of archaea, with higher values reflecting increased contributions from archaeal communities residing at deeper water depths (Rattanasriampaipong et al., 2022). Specifically, [2]/[3] values above 5 indicate a significant contribution from deeper-dwelling archaea (Taylor et al., 2013). While PC1 for all global data shows only a weak correlation with [2]/[3] (Table 2; R = 0.38), PC1 for Southern ocean data exhibits a much stronger correlation (Table 2; R = 0.74) with a steeper slope, including several values exceed this threshold (Fig. S4). This association implies that at some sites sedimentary isoGDGTs in the Southern Ocean have a stronger contribution from archaea living in the deeper ocean.

The spatial distribution of [2]/[3] exhibits a latitudinal pattern: lower values occur south of the ACC, whereas northern/lower-latitude sites frequently have higher values exceeding the threshold (> 5) (Fig. S3). If isoGDGT distributions were controlled solely by temperature, archaea at lower-latitude sites would produce GDGTs with more rings, leading to lower [2]/[3] values. However, the pattern we observe here appears counterintuitive, with northern sites clustering at higher values.

Several mechanisms may explain how water mass structure affects [2]/[3] values. First, basin bathymetry provides control on [2]/[3] distributions. Our finding that [2]/[3] ratios correlate with site water depth (R = 0.74) suggests that in deeper basins the relative contribution from deep-dwelling archaea is increased. This bathymetric effect is particularly pronounced at northern Southern Ocean sites, where core-top samples predominantly come from deep basin locations (mean depth ~3400 m). Conversely, sites on the shallower Antarctic continental shelf and slope maintain lower [2]/[3] ratios likely due to shallow archaeal clades dominating the water column.

Additionally, the relationship between export depth and the depth of maximum isoGDGT production differs within the water column. Peak isoGDGT production is thought to occur at the base of the photic zone and/or below the nitracline (Hernández-Sánchez et al., 2014). Nitracline depths in the Southern Ocean vary with latitude, with northern sites, particularly in the South Pacific subtropical gyre region, exhibiting deeper nitraclines (several hundred meters to 1000 m) due to strong permanent stratification and Ekman downwelling at gyre centres (Dai et al., 2023; Feucher et al., 2019). These deeper nitraclines could increase contributions from deeper archaeal clades to seafloor sediment. In addition, the oligotrophic conditions at northern subtropical sites result in low surface particle flux (Dai et al., 2023). Since export of isoGDGTs depends on sinking particles (e.g., marine snow, fecal pellets, phytoplankton aggregates) that incorporate lipids (Huguet et al., 2006; Wuchter et al., 2005), weak surface productivity reduces the contribution of surface-dwelling archaeal signals, allowing GDGTs from deeper archaeal communities to contribute proportionally more to the sedimentary record, thereby elevating [2]/[3] values. In contrast, southern sites are characterized by upwelling of Circumpolar Deep Water, which brings nutrients to the upper water column and creates shallower nitraclines with enhanced surface productivity (Carter et al., 2008), leading to lower [2]/[3] values. These mechanisms can potentially explain the stronger PC1 and [2]/[3] correlation observed in the Southern Ocean compared to globally, introducing a significant non-thermal influence on sedimentary isoGDGT distributions across frontal boundaries."

**L239: the discussion suddenly shifts to differences in isoGDGTs north and south of the polar front, whereas these relations have not been shown or discussed yet. This only happens much later in the manuscript. Critically check the story line and adjust the order.**
Reply: We thank the reviewer for this feedback. We will adjust the order of when we explain these concepts, so they are brought in earlier in the manuscript.

**AIZ index: it is not clear to me how one can distinguish a water mass signal vs a temperature signal from this index. Please elaborate. This is important given your finding that the AIZ index relates to temperature south of the PF. When applying this index downcore, it has to be clear how this record needs to be interpreted. Are we reconstructing the position of the PF? Or temperature? And can the temperature relationship be extrapolated (as suggested in L329), or is it only valid <4C/in sediments below water masses south of the PF? What does the record mean in terms of temperature when the AIZ index exceeds the PF threshold value over time?**
Reply: We thank the reviewer for this feedback. As we addressed in response to previous comments, we propose that because isoGDGTs in the region south of the PF are predominantly sourced from *Nitrosopumilales*, the temperature sensitivity of the index south of the PF (when AIZ values are below 0.14 ± 0.03 SD) is robust. We will also add the following explanation in "Section 4.5 Regional core-top calibration for AIZ index" to clarify the validity of the AIZ index as a temperature proxy:

L334: "It is important to note that AIZ index-based palaeothermometry is only applicable within the Antarctic Zone, as indicated by AIZ values below 0.14 (± 0.03 SD), a threshold based on the present PF position. If a sediment core reconstruction contains intervals where AIZ values are both higher and lower than 0.14, then temperatures should only be reconstructed for intervals with values lower than 0.14, with 0.14 used a marker for when the PF has crossed the core site."

**In section 4.6.2 should be made more clear that the temperature discussion is only valid for locations south of the PF. Also for the comparison with OPTIMAL: the record from north of the PF does not show clear G-IG cycles as suggested in the text. Please adjust.**
Reply: We will revise section 4.6.2 accordingly by adding explicit clarification that the temperature discussion and AIZ-based reconstructions are only valid for locations south of the PF. As we addressed in response to the previous comment, we will incorporate the threshold criterion (AIZ < 0.14) and emphasize that this spatial constraint ensures the reliability of temperature estimates within the south of the PF (*Nitrosopumilales*-dominated) region.

Reply: Regarding the OPTIMAL record from north of the PF, we acknowledge that the MD11-3357 record does not show clear glacial-interglacial (G-IG) cycles. We will revise the text to: "L395: OPTIMAL-derived SST records at most sites (except MD11-3357) represent typical G-IG cycles..."

**There are a lot of supplementary figures. Are they all necessary?**
Reply: We appreciate this suggestion. We will remove supplementary Figures S5, S6, S9, and S10, which overlap with other figures or are insufficiently discussed in the main text. This will reduce the supplementary figures from 10 to 6 in the revised manuscript
* * *
*Additional references (not found in our original submission) referred to by the reviewer and in our response – these will be incorporated into our final revised submission:*

Alonso-Sáez, L., Andersson, A., Heinrich, F., and Bertilsson, S.: High archaeal diversity in Antarctic circumpolar deep waters: Biogeography of Archaea in Antarctic waters, Env. Microbiol. Rep., 3, 689–697, https://doi.org/10.1111/j.1758-2229.2011.00282.x, 2011.

Bale, N. J., Palatinszky, M., Irene, W., Rijpstra, C., Herbold, C. W., Wagner, M., Sinninghe Damsté, J. S., and Atomi, H.: Membrane Lipid Composition of the Moderately Thermophilic Ammonia-Oxidizing Archaeon "Candidatus Nitrosotenuis uzonensis" at Different Growth Temperatures, Appl Environ Microbiol, 85, e01332-19, https://doi.org/10.1128/AEM.01332-19, 2019.

Bijl, P. K., Śliwińska, K. K., Duncan, B., Huguet, A., Naeher, S., Rattanasriampaipong, R., Sosa-Montes De Oca, C., Auderset, A., Berke, M. A., Kim, B. S., Davtian, N., Dunkley Jones, T., Eefting, D. D., Elling, F. J., Fenies, P., Inglis, G. N., O'Connor, L., Pancost, R. D., Peterse, F., Rice, A., Sluijs, A., Varma, D., Xiao, W., and Zhang, Y. G.: Reviews and syntheses: Best practices for the application of marine GDGTs as proxy for paleotemperatures: sampling, processing, analyses, interpretation, and archiving protocols, Biogeosciences, 22, 6465–6508, https://doi.org/10.5194/bg-22-6465-2025, 2025.

Dai, M., Luo, Y., Achterberg, E. P., Browning, T. J., Cai, Y., Cao, Z., Chai, F., Chen, B., Church, M. J., Ci, D., Du, C., Gao, K., Guo, X., Hu, Z., Kao, S., Laws, E. A., Lee, Z., Lin, H., Liu, Q., Liu, X., Luo, W., Meng, F., Shang, S., Shi, D., Saito, H., Song, L., Wan, X. S., Wang, Y., Wang, W., Wen, Z., Xiu, P., Zhang, J., Zhang, R., and Zhou, K.: Upper Ocean Biogeochemistry of the Oligotrophic North Pacific Subtropical Gyre: From Nutrient Sources to Carbon Export, Reviews of Geophysics, 61, e2022RG000800, https://doi.org/10.1029/2022RG000800, 2023.

Feucher, C., Maze, G., and Mercier, H.: Subtropical Mode Water and Permanent Pycnocline Properties in the World Ocean, JGR Oceans, 124, 1139–1154, https://doi.org/10.1029/2018JC014526, 2019.

Hernández, E. A., Piquet, A. M.-T., Lopez, J. L., Buma, A. G. J., and Mac Cormack, W. P.: Marine archaeal community structure from Potter Cove, Antarctica: high temporal and spatial dominance of the phylum Thaumarchaeota, Polar Biol, 38, 117–130, https://doi.org/10.1007/s00300-014-1569-8, 2015.

Kim, J.-G., Park, S.-J., Quan, Z.-X., Jung, M.-Y., Cha, I.-T., Kim, S.-J., Kim, K.-H., Yang, E.-J., Kim, Y.-N., Lee, S.-H., and Rhee, S.-K.: Unveiling abundance and distribution of planktonic Bacteria and Archaea in a polynya in Amundsen Sea, Antarctica, https://doi.org/10.1111/1462-2920.12287, 2014.

Kolody, B. C., Sachdeva, R., Zheng, H., Füssy, Z., Tsang, E., Sonnerup, R. E., Purkey, S. G., Allen, E. E., Banfield, J. F., and Allen, A. E.: Overturning circulation structures the microbial functional seascape of the South Pacific, Science, 2025.

Pancost, R. D., Hopmans, E. C., and Sinninghe Damsté, J. S.: Archaeal lipids in Mediterranean cold seeps: molecular proxies for anaerobic methane oxidation, Geochimica et Cosmochimica Acta, 65, 1611–1627, https://doi.org/10.1016/S0016-7037(00)00562-7, 2001.

Raes, E. J., Bodrossy, L., Van De Kamp, J., Bissett, A., Ostrowski, M., Brown, M. V., Sow, S. L. S., Sloyan, B., and Waite, A. M.: Oceanographic boundaries constrain microbial diversity gradients in the South Pacific Ocean, Proc. Natl. Acad. Sci. U.S.A., 115, https://doi.org/10.1073/pnas.1719335115, 2018.

Rattanasriampaipong, R., Zhang, Y. G., Pearson, A., Hedlund, B. P., and Zhang, S.: Archaeal lipids trace ecology and evolution of marine ammonia-oxidizing archaea, Proc. Natl. Acad. Sci. U.S.A., 119, e2123193119, https://doi.org/10.1073/pnas.2123193119, 2022.

Santoro, A. E., Dupont, C. L., Richter, R. A., Craig, M. T., Carini, P., McIlvin, M. R., Yang, Y., Orsi, W. D., Moran, D. M., and Saito, M. A.: Genomic and proteomic characterization of " *Candidatus* Nitrosopelagicus brevis": An ammonia-oxidizing archaeon from the open ocean, Proc. Natl. Acad. Sci. U.S.A., 112, 1173–1178, https://doi.org/10.1073/pnas.1416223112, 2015.

Varma, D., Hopmans, E. C., Van Kemenade, Z. R., Kusch, S., Berg, S., Bale, N. J., Sangiorgi, F., Reichart, G.-J., Sinninghe Damsté, J. S., and Schouten, S.: Evaluating isoprenoidal hydroxylated

GDGT-based temperature proxies in surface sediments from the global ocean, Geochim. Cosmochim. Acta, 370, 113–127, https://doi.org/10.1016/j.gca.2023.12.019, 2024.

Wuchter, C., Schouten, S., Wakeham, S. G., and Sinninghe Damsté, J. S.: Temporal and spatial variation in tetraether membrane lipids of marine Crenarchaeota in particulate organic matter: Implications for TEX$_{86}$ paleothermometry, https://doi.org/10.1029/2004PA001110, 2005.

Zhao, S., Bao, R., Zhou, L., Liu, M., Sun, X., Gao, Z., Zhou, Y., Wang, N., Wang, Y., Chen, J., Xing, L., and Zhang, C.: Temperature-dependent spatial and temporal trends in archaeal lipid distributions, Commun Earth Environ, 6, 619, https://doi.org/10.1038/s43247-025-02450-7, 2025.